# GPTAraEval: A Comprehensive Evaluation of ChatGPT on Arabic NLP

**Md Tawkat Islam Khondaker**[λ]      **Abdul Waheed**[ξ]

**El Moatez Billah Nagoudi**[λ]      **Muhammad Abdul-Mageed**[λ,ξ]

[λ] Deep Learning & Natural Language Processing Group, The University of British Columbia

[ξ]Department of Natural Language Processing & Department of Machine Learning, MBZUAI

{tawkat@cs,moatez.nagoudi,muhammad.mageed}@ubc.ca

## Abstract

ChatGPT's emergence heralds a transformative phase in NLP, particularly demonstrated through its excellent performance on many English benchmarks. However, the model's efficacy across diverse linguistic contexts remains largely uncharted territory. This work aims to bridge this knowledge gap, with a primary focus on assessing ChatGPT's capabilities on Arabic languages and dialectal varieties. Our comprehensive study conducts a large-scale automated and human evaluation of ChatGPT, encompassing 44 distinct language understanding and generation tasks on over 60 different datasets. To our knowledge, this marks the first extensive performance analysis of ChatGPT's deployment in Arabic NLP. Our findings indicate that, despite its remarkable performance in English, ChatGPT is consistently surpassed by smaller models that have undergone finetuning on Arabic. We further undertake a meticulous comparison of ChatGPT and GPT-4's Modern Standard Arabic (MSA) and Dialectal Arabic (DA), unveiling the relative shortcomings of both models in handling Arabic dialects compared to MSA. Although we further explore and confirm the utility of employing GPT-4 as a potential alternative for human evaluation, our work adds to a growing body of research underscoring the limitations of ChatGPT.

## 1 Introduction

Large language models (LLMs) pretrained on next token prediction brought significant progress to NLP. These models can be finetuned to follow human instructions, allowing users to steer model output (Wei et al., 2021; Wu et al., 2021; Chung et al., 2022b; Ouyang et al., 2022; Muennighoff et al., 2022a). ChatGPT[1] is the most prominent among these models and has recently received significant

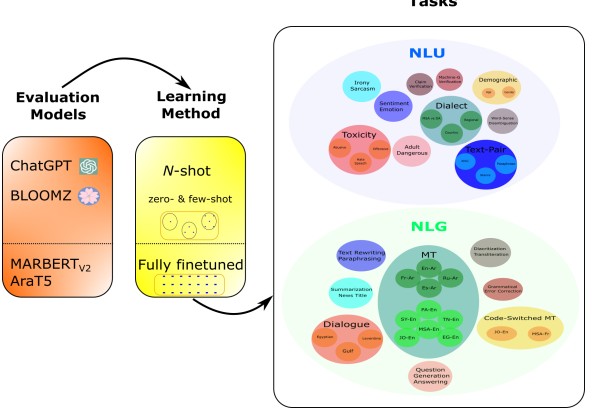

Figure 1: Experimental setup for our evaluation. We evaluate ChatGPT on 44 Arabic NLP tasks.

attention due to its remarkable abilities. [2]

The underlying model behind ChatGPT is believed to be pretrained on large datasets from multiple languages, which could enable the model to work well in multilingual settings. There is, however, much hype about ChatGPT's abilities. In fact, some observations around ChatGPT remain anecdotal, calling for a rigorous evaluation of its abilities in different languages. While multiple investigations of ChatGPT performance on English are accumulating fast (Qin et al., 2023a; Gilardi et al., 2023; Laskar et al., 2023), evaluation on other languages remains largely speculative. This lack of methodic evaluation inhibits our understanding of the capabilities and limitations of LLMs beyond English. This motivates our work for evaluating ChatGPT on the wide collection of languages and language varieties commonly referred to as Arabic. Arabic has rich morphologies and syntactic structures. It also has a vast native speaker population of over 450 million people, making investigations of it necessary for *technological inclusion*. Combined with this large population, the complexity of Arabic

---

[1]https://openai.com/blog/chatgpt

[2]In this paper, we use the term ChatGPT to refer to the *gpt-3.5-turbo-0301* model.

requires models to have a deep understanding of expansive sociolinguistic contexts. This also makes the evaluation on Arabic all the more important to our understanding of how large multilingual models behave, with implications that go *well beyond* Arabic.

Concretely, our objective is to systematically evaluate ChatGPT on a wide range of Arabic natural language understanding (NLU) and natural language generation (NLG) tasks, identifying existing gaps and ultimately possibly guiding the development of more effective Arabic and multilingual applications. Through our comprehensive benchmarking on 44 diverse tasks comprising over 60 datasets, we observe that ChatGPT exhibits inferior performance compared to much smaller finetuned models (Section 6 and Section 7). When we consider the performance of ChatGPT and GPT-4 on country-level dialectal Arabic as compared to Modern Standard Arabic (MSA), the modern-day variety used in formal settings, we reveal even more inferior dialectal performance (Section 8). To the best of our knowledge, this is the first work to study the comparison of ChatGPT and GPT-4 on Arabic dialectal variation.

The present study unambiguously demonstrates that, notwithstanding its noteworthy performance on standardized English benchmark datasets (Zhong et al., 2023a), the efficacy of ChatGPT as a universally applicable solution appears to be unsubstantiated when considering languages characterized by extensive linguistic variations, particularly in the context of Arabic. Succinctly put, it is evident that considerable enhancements can still be made pertaining to Arabic and other linguistically analogous languages. Our contributions can be summarized as follows:

1. We rigorously evaluate ChatGPT on a wide range of Arabic tasks and varieties, offering the first work benchmarking ChatGPT on Arabic NLP at scale.

2. We contextualize ChatGPT by comparing it against BLOOMZ, another relatively large model (**7.1**B), and two finetuned dedicated Arabic models.

3. We perform a systematic evaluation of ChatGPT and GPT-4 on dialectal Arabic (DA) as compared to MSA, revealing the weakness of both models on especially dialects.

4. We further conduct quality assessments on the models' generations using both human and GPT-4 as evaluators, finding GPT-4 evaluation to notably align with human evaluation.

5. Through our empirical analyses, we find that ChatGPT significantly lags behind much smaller Arabic-focused finetuned models on almost all tasks. Our findings should motivate future work focused at improving LLM performance on Arabic languages and varieties.

## 2 Related Work

We provide a brief overview of the previous works that evaluate ChatGPT on NLP tasks here. We offer a detailed walkthrough in Appendix A.

**Performance on English.** In *machine translation (MT)*, while ChatGPT equates to commercial tools like Google Translate (Jiao et al., 2023), it falls short in domain-specific translation and low-resource languages. However, strategies like pivot prompting (Jiao et al., 2023) and additional information-infused prompts (Peng et al., 2023; Gao et al., 2023) can enhance performance. In *question-answering (QA)*, while ChatGPT shows potential (Zheng et al., 2023; Shen et al., 2023; Omar et al., 2023), it falters on complex open-domain questions and tasks requiring reasoning (Zheng et al., 2023; Tan et al., 2023) and is vulnerable to adversarial examples and perturbations (Shen et al., 2023). For *text classification*, ChatGPT performs well in zero-shot and few-shot settings (Zhong et al., 2023b; Gilardi et al., 2023; Wang et al., 2023b), sometimes matching or surpassing fully supervised models.

**Performance on Multilingual Tasks.** In multilingual tasks, ChatGPT is found to struggle with generating non-Latin scripts (Bang et al., 2023), but strategies such as prompting task descriptions in a high-resource language such as English can improve results (Lai et al., 2023). In line with this research, we find ChatGPT to work better with English prompts than Arabic prompts. Lai et al. (2023) evaluate ChatGPT in zero-shot setting on seven tasks covering 37 languages including Arabic. The authors show ChatGPT performs generally well for high-resource languages such as English, Russian, and German compared to medium-resource languages such as Arabic. Huang et al. (2023) introduce cross-lingual thought and assess

the multilingual capability of ChatGPT on 7 benchmarks. The authors specifically evaluate ChatGPT on Arabic for natural language inference task in few-shot settings. Compared to these concurrent efforts, our work conducts ChatGPT evaluation on Arabic at a much larger scale with 44 diverse tasks, as well as a comprehensive analysis of ChatGPT and GPT-4's performance on dialectal varieties. Wu et al. (2023a) find that ChatGPT fails to surpass the passing score in Chinese medical licensing exam. Similarly, Kasai et al. (2023) show that ChatGPT fails to pass the Japanese medical licensing exam. In our work, we conclusively show that ChatGPT is inferior to supervised models across a wide range of domains on several varieties of Arabic.

## 3 Datasets

We evaluate ChatGPT on both Arabic NLU and NLG tasks. For NLU, we use a total of 48 datasets representing 20 different tasks from the ORCA benchmark (Elmadany et al., 2023). The diverse coverage of these datasets lends our evaluation broad representativeness and applicability to many real-world scenarios. For NLG, we compile 23 publicly available datasets from across 13 task clusters. We briefly introduce our evaluation datasets below and provide a detailed description of them in Appendix B.

**NLU Tasks.** In NLU, we include 20 tasks spanning across Arabic-NLI (1),[3] claim prediction (1), dialect identification (3), machine-generated text detection (1), paraphrase detection (1), sentiment analysis and emotion detection (2), social meaning detection (9), stance detection (1), and word sense disambiguation (1). Task-specific details with full citation of all datasets are in Appendix B.1.

**NLG Tasks.** We include 24 NLG tasks covering 13 task clusters: code-switched translation (2), diacritization (1), dialect translation (6), grammatical error correction (1), MT (4), news title generation (1), open-domain dialectal generation (3), paraphrasing (1), QA (1), question generation (1), text rewriting (1), transliteration (1), and summarization (1). Again, we provide task-specific details with appropriate references in Appendix B.2.

## 4 Prompt Design

A prompt is a set of instructions provided to an LLM that programs the model by enhancing its purpose and capabilities (White et al., 2023). A

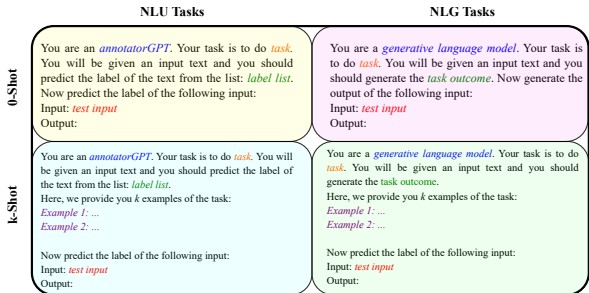

Figure 2: Prompt templates for different tasks.

prompt can influence subsequent interactions with the model as well as its generated outputs by defining a set of specific rules. Therefore, in order to acquire a desired outcome, it is important to clearly define the set of rules and intents to the model. In our initial experiments, we experimented with a diverse set of prompt templates in both English and Arabic. Specifically, we evaluated on dialect identification, machine-generated text detection, and toxicity detection for NLU and machine translation tasks for NLG with both English and Arabic prompts, observing that the English prompt outperforms its Arabic counterpart. Hence, we design a universal prompt template in English as follows: **(1)** We first set the *role* of the model (e.g., as an annotator for NLU tasks). **(2)** We provide name of the *task* that needs to be performed (e.g., sentiment classification). **(3)** We define what should be the expected *outcome* of the model (e.g., the label set for an NLU task). **(4)** If the task involves $k$-shot ($k > 0$) learning, we provide the model with $k$ *examples*. **(5)** Finally, we deliver *test input* to the model to produce the output. We present the templates of our designed prompts in Figure 2. We offer examples of $k$-shot NLU and NLG tasks in Appendix H.

## 5 Experiments

We run experiments under 0-shot, 3-shot, 5-shot, and 10-shot settings. For the training data of the few-shot experiments, we randomly sample from each respective training dataset.[4] For a $k$-shot setting, we make sure that if a training sample is selected then it will also be selected for $n$-shot settings where $n > k$. For evaluation, we randomly sample a set of 200 examples from the test set of each dataset, to keep the cost man-

---

[3]Number of tasks in each cluster are in parentheses.

---

[4]Comparison between the class distribution of the whole training set and the few-shot samples is in Appendix F.

ageable. We evaluate ChatGPT (gpt-3.5-turbo),[5] which is an optimized version of GPT-3.5 series. We set the temperature to 0.0 while generating responses from ChatGPT. We compare this model with BLOOMZ (7.1B parameters),[6] which is fine-tuned on a diverse set of tasks on 46 languages including Arabic (Muennighoff et al., 2022b). We choose BLOOMZ since it is claimed to achieve impressive generalization on unseen tasks. Following Chung et al. (2022a), we use free-form generation for both ChatGPT and BLOOMZ along with simple post-processing, e.g. removing leading or trailing whitespace. Additionally, to further situate performance of ChatGPT, we fine-tune MARBERT$_{V2}$ (Abdul-Mageed et al., 2021a) and AraT5 model (Nagoudi et al., 2022b). Again, we choose these models as they are reported to achieve SOTA on NLU and NLG in Abdul-Mageed et al. (2021a) and Nagoudi et al. (2022b), respectively. For both MARBERT$_{V2}$ and AraT5, we identify the best model on the respective development split (Dev). We train MARBERT$_{V2}$ for 25 epochs with a patience of 5 and AraT5 for 10 epochs. For both models, we set the learning rate to $5e$-5 across all the tasks. We present our overall experimental setup in Figure 1. In addition to reporting the performance of finetuned models on the same 200 set as ChatGPT and BLOOMZ, we report the 3-run average of the models' performances on the respective full test sets (reported in gray). For NLU tasks, we use macro-F$_1$ scores and for NLG tasks we use the appropriate metric suited to each task (Table 2). We provide examples of NLG model responses in Appendix I.

## 6  Evaluation on NLU Tasks

**Overview.** We present our evaluation on NLU tasks in Table 1. We observe that *although* ChatGPT *outperforms instruction-tuned multilingual models like* BLOOMZ, *the smaller finetuned model* MARBERT$_{V2}$ *achieves significantly better results than* ChatGPT. We also find that model performance does not necessarily increase as we increase the number of shots. The common wisdom for this behavior is that improvement with few-shot learning is model- and task-dependent, and is often sensitive to the order of the shots Wei et al. (2021); Brown et al. (2020); Lu et al. (2022). We now discuss performance on each understanding task.

**Emotion and Sentiment.** ChatGPT achieves 17.33% F$_1$ on emotion detection with 5-shot learning, while BLOOMZ achieves 17.13 with 10-shot. Both of these models, however, perform much lower than finetuned MARBERT$_{V2}$ (F$_1$=68.74). For sentiment, ChatGPT (64.96)[7] outperforms BLOOMZ (50.44) by a margin of 14.52 F$_1$ scores. Again, MARBERT$_{V2}$ (F$_1$=68.92) outperforms both models.

**Dialect.** On *binary-level* dialect classification, ChatGPT achieves 77.00 F$_1$ with 3-shot learning, outperforming BLOOMZ (49.68 F$_1$) by 27.32 points. Meanwhile, MARBERT$_{V2}$ (88.00) significantly outperforms both models. On *region-level* dialect, ChatGPT achieves 30.69 F$_1$, outperforming BLOOMZ (30.19). MARBERT$_{V2}$ outperforms both models (89.69). On *country-level* dialect, BLOOMZ exhibits lower performance (14.61) compared to ChatGPT (19.74) and MARBERT$_{V2}$ (33.88). Similar to the two other dialect tasks, ChatGPT falls behind the much smaller model on country-level dialect ID.

**Claim and Machine-Generated Text.** For claim verification, performance of ChatGPT increases with the increased number of shots (achieving 61.82 F$_1$). BLOOMZ scores less than half (30.87) compared to ChatGPT. ChatGPT falls behind MARBERT$_{V2}$ by a margin of 3.71. On machine-generated text detection, ChatGPT and BLOOMZ achieve comparable performance (47.81 and 45.83, respectively). MARBERT$_{V2}$ (76.23) outperforms both models by a large margin.

**Toxic Text.** On *abusive language detection*, the best score ChatGPT achieves is 42.03% (with 5-shot learning). This is close to BLOOMZ (42.26, with 5-shot). Again, MARBERT$_{V2}$ is much better than both (82.96). On *hate speech detection*, ChatGPT achieves 50.99. It is outperformed by BLOOMZ, which acquires 57.88 with 10-shot learning. Both models are significantly outperformed by MARBERT$_{V2}$ (78.95). On *offensive language*, ChatGPT is at 73.76 (with 10-shots), and even its 0-shot outperforms all few-shots of BLOOMZ. Similar to other tasks, MARBERT$_{V2}$ performs best (97.37). Given concerns about models like ChatGPT generating toxic language, we also *manually* inspect its errors finding it to be sensitive to false toxicity (i.e., it tends to flag non-toxic text as toxic). Refer to Appendix D for our full toxicity analysis.

**Irony and Sarcasm.** On *irony* detection, 0-shot ChatGPT achieves 67.27, outperforming BLOOMZ

[5]Snapshot of gpt-3.5-turbo from March 1st 2023.
[6]https://huggingface.co/bigscience/BLOOMZ-7b1

[7]When we report a result without reference to the number of shots, it is typically the best result across all shots.

| Task | BLOOMZ (N-shot) | | | | ChatGPT (N-shot) | | | | MARBERT$_{V2}$ (Test No.) | MARBERT$_{V2}$ (Test No.) |
|---|---|---|---|---|---|---|---|---|---|---|
| | 0 | 3 | 5 | 10 | 0 | 3 | 5 | 10 | 200 | Full |
| Dialect-Binary | 7.76 | 47.81 | 44.47 | 49.68 | 66.98 | 77.00 | 70.33 | 70.81 | **88.00** | 86.91 |
| Dialect-Region | 2.31 | 20.28 | 30.19 | 22.50 | 30.69 | 29.52 | 27.98 | 27.13 | **89.69** | 66.32 |
| Dialect-Country | 0.81 | 12.34 | 12.84 | 14.61 | 13.86 | 18.60 | 19.74 | 19.43 | **33.88** | 36.06 |
| Machine-Gen | 2.64 | 29.08 | 45.83 | 45.83 | 26.96 | 31.90 | 19.50 | 47.81 | **76.23** | 86.69 |
| Abusive | 13.21 | 23.24 | 42.26 | 42.10 | 36.90 | 34.98 | 42.03 | 36.60 | **82.96** | 78.03 |
| Hate speech | 9.01 | 52.61 | 51.06 | 57.88 | 37.28 | 50.99 | 44.79 | 47.38 | **78.95** | 83.54 |
| Offensive | 12.78 | 60.01 | 47.48 | 37.94 | 60.32 | 71.61 | 69.34 | 73.76 | **97.37** | 92.23 |
| Irony | 13.44 | 54.81 | 49.66 | 52.51 | 67.27 | 52.51 | 59.51 | 57.37 | **85.83** | 83.09 |
| Sarcasm | 2.76 | 30.94 | 28.28 | 29.41 | 42.58 | 41.71 | 43.99 | 43.09 | **80.10** | 76.19 |
| Dangerous | 45.07 | 17.40 | 22.48 | 18.97 | 24.35 | 27.98 | 26.97 | 37.11 | **63.97** | 67.11 |
| Adult | 3.77 | 63.70 | 57.81 | 52.15 | 41.41 | 62.32 | 62.36 | 66.12 | **93.95** | 90.97 |
| Gender | 7.14 | 40.48 | 35.52 | 36.32 | 1.62 | 8.18 | 27.36 | **65.04** | 64.97 | 67.64 |
| Age | 1.05 | 27.27 | 29.64 | 21.80 | 10.05 | 35.58 | 42.21 | 44.11 | **49.41** | 46.24 |
| Claim | 25.93 | 25.72 | 25.86 | 30.87 | 24.09 | 54.24 | 56.02 | 61.82 | **65.53** | 67.83 |
| Emotion | 11.40 | 9.66 | 15.84 | 17.13 | 14.54 | 17.23 | 17.33 | 16.58 | **68.74** | 70.82 |
| Sentiment | 43.72 | 45.53 | 50.00 | 50.44 | 58.00 | 62.37 | 64.96 | 63.18 | **68.92** | 80.83 |
| Paraphrase | 34.64 | 62.09 | 63.66 | 48.93 | 42.16 | 82.45 | 82.97 | **85.80** | 65.98 | 63.47 |
| Stance | 14.53 | 23.85 | 24.49 | 21.66 | 61.51 | 62.22 | 66.60 | 63.70 | **88.23** | 80.57 |
| XNLI | 17.05 | 17.28 | 17.48 | 26.32 | 56.43 | 50.87 | 49.40 | 55.66 | **62.89** | 62.22 |
| WSD | 43.75 | 41.36 | 46.43 | 47.11 | 42.50 | **53.49** | 43.96 | 43.12 | 36.31 | 33.28 |
| **ORCA**$_{Score}$ | 15.64 | 35.27 | 37.06 | 36.21 | 37.98 | 46.29 | 46.87 | 51.28 | **68.85** | 71.00 |

Table 1: NLU Results. We report the Macro-F$_1$ score for every task. We evaluate BLOOMZ and ChatGPT in 0-shot and in-context n-shot (where n = 3, 5, 10) settings. MARBERT$_{V2}$ is our fully supervised model. We report the results of MARBERT$_{V2}$ on the same test set (200 samples) as BLOOMZ and ChatGPT for fair comparison. The best scores are in **bold**. We additionally report the performance of MARBERT$_{V2}$ on the full test set from Elmadany et al. (2022).

but again lags behind MARBERT$_{V2}$ (85.83) by a margin of 18.56 points. On *sarcasm*, ChatGPT (43.99) outperforms BLOOMZ (30.94) but is almost half of MARBERT$_{V2}$ performance (80.10).

**Adult and Dangerous Content.** ChatGPT achieves 66.12 F$_1$ with 10-shot learning on *adult content detection*, while BLOOMZ achieves at best 63.70 (with 3-shots). Aligning with the general trend thus far, MARBERT$_{V2}$ (93.95) outperforms both models by a significant margin. On the *dangerous content* dataset, ChatGPT only achieves 37.11 F$_1$. Interestingly, ChatGPT is outperformed by BLOOMZ (45.07) in 0-shot learning. However, ChatGPT dominates BLOOMZ in all other few-shot setups. Again, MARBERT$_{V2}$ outperforms both models (63.97).

**Demographic Text Classification.** On *age* prediction, ChatGPT achieves 44.11 F$_1$ (with 10-shots), whereas BLOOMZ is at 29.64 (with 5-shots). Here, MARBERT$_{V2}$ (49.41) outperforms ChatGPT by 5.30. On the *gender* task, ChatGPT (65.04) outperforms BLOOMZ (40.48). Also, ChatGPT performs better than MARBERT$_{V2}$ (64.97) by a slight margin.

**Word Sense Disambiguation.** ChatGPT achieves the best score of 53.49 with 3-shots. The other few-shot settings of ChatGPT are outperformed by the corresponding few-shot settings of BLOOMZ. Surprisingly, the finetuned model MARBERT$_{V2}$ is outperformed by both ChatGPT and BLOOMZ by a significant margin. We suspect this is due to the issue of *anisotropy* (Ethayarajh, 2019; Li et al., 2020)

in BERT models, which we further discuss in Appendix C.

**Text-Pair Tasks.** On *paraphrase identification*, ChatGPT with 10-shots (85.80) outperforms both BLOOMZ (63.66 with 5-shot) and MARBERT$_{V2}$ (65.98) by a large margin. Strikingly, ChatGPT with only 3- or 5-shots outperforms the fully-finetuned MARBERT$_{V2}$ model. This shows the remarkable ability of ChatGPT on semantic tasks such as paraphrase detection. On *stance* detection, all the few-shot setups of ChatGPT outperform BLOOMZ by a significant margin. However, MARBERT$_{V2}$ (F$_1$=88.23) outperforms ChatGPT. On natural language inference (XNLI), ChatGPT achieves F$_1$ of 56.43 with 0-shot learning, while the best score of BLOOMZ is at 26.32 (for 10-shots). Both models are outperformed by MARBERT$_{V2}$ (62.89).

## 7  Evaluation on NLG Tasks

**Overview.** We present the results of our evaluation on generation tasks in Table 2. For NLG, we notice that ChatGPT *performs better than* BLOOMZ *on the majority of tasks. However, following a similar trend to NLU, finetuned* AraT5 *consistently outperforms* ChatGPT. We now present our results for each group of NLG tasks.

**Text Rewriting and Paraphrase.** For text rewriting, BLOOMZ achieves 76.67 BLEU scores (with 0-shots) which is better than ChatGPT results. Surprisingly, BLOOMZ's performance deteriorates as we

| Task | Metric | BLOOMZ (N-shot) | | | | ChatGPT (N-shot) | | | | AraT5 (Test No.) | AraT5 (Test No.) |
|------|--------|---|---|---|----|---|---|---|----|---|---|
| | | 0 | 3 | 5 | 10 | 0 | 3 | 5 | 10 | 200 | Full |
| Text Rewriting | BLEU | 76.67 | 23.96 | 13.97 | 12.73 | 41.59 | 58.75 | 53.34 | 62.62 | **99.64** | 91.19 |
| Paraphrase | BLEU | 12.98 | 9.37 | 10.27 | 10.55 | 7.89 | 8.92 | 9.19 | 9.60 | **14.40** | 18.69 |
| Question-Gen | BLEU | 28.76 | 15.70 | 18.53 | 18.69 | 14.48 | 19.86 | 20.08 | 18.15 | **35.17** | 33.64 |
| QA | SQuAD $F_1$ | 76.04 | 65.45 | 62.08 | 60.49 | 32.98 | 51.73 | 54.14 | 53.67 | **81.45** | 83.34 |
| Summarization | ROUGE-L | 13.56 | 9.13 | 10.74 | 9.63 | 16.88 | 20.01 | 20.43 | 19.58 | **35.31** | 26.88 |
| News Title-Gen | BLEU | 0.99 | 0.79 | 1.20 | 0.62 | 3.24 | 4.72 | 4.62 | 4.54 | **7.72** | 9.64 |
| Diacritization ↓ | CER | 0.51 | 1.33 | 1.62 | 1.42 | 0.11 | 0.06 | 0.05 | 0.06 | **0.03** | 0.01 |
| Transliteration ↓ | CER | 0.59 | 0.45 | 0.42 | 0.42 | 0.27 | 0.24 | 0.24 | 0.23 | **0.18** | 0.18 |
| MT (en→ar) | BLEU | 8.33 | 12.54 | 12.35 | 10.07 | 20.52 | 23.58 | 23.34 | 23.74 | **27.12** | 28.12 |
| MT (es→ar) | BLEU | 6.94 | 9.20 | 9.31 | 7.33 | 16.47 | 18.11 | 17.45 | 19.32 | **21.16** | 21.74 |
| MT (fr→ar) | BLEU | 6.88 | 5.51 | 5.76 | 4.97 | 15.12 | 15.44 | 15.57 | 16.26 | **18.48** | 20.51 |
| MT (ru→ar) | BLEU | 2.42 | 1.95 | 3.17 | 1.82 | 15.83 | 17.52 | 17.46 | 17.38 | **19.32** | 18.29 |
| CST (Jo-en→en) | BLEU | 11.52 | 10.91 | 11.56 | 11.50 | 36.61 | 37.38 | 38.55 | **40.88** | 5.56 | 6.29 |
| CST (Dz-fr→fr) | BLEU | 28.41 | 28.27 | 26.75 | 28.61 | 34.61 | 35.40 | 36.45 | **37.95** | 17.49 | 16.16 |
| GEC | $M^2$Scorer $F_{0.5}$ | 2.40 | 1.42 | 2.40 | – | 48.72 | 48.72 | 46.56 | – | **67.54** | 70.54 |

Table 2: NLG Results. Higher is better unless otherwise specified by ↓. We evaluate BLOOMZ and ChatGPT in 0-shot and in-context n-shot (where n = 3, 5, 10) settings. AraT5 is our fully supervised model. The best scores are in **bold**. **QA** - Question Answering, **MT** - Machine Translation, **CST** - Code Switched Translation. We report the results of AraT5 on the same test set (200 samples) as BLOOMZ and ChatGPT for a fair comparison.

increase the number of training examples whereas ChatGPT shows best performance (62.62) with 10-shots. However, both models are significantly dominated by AraT5. For paraphrase generation, BLOOMZ outperforms ChatGPT in all $k$-shot setups. Noticeably, the performance of ChatGPT monotonically improves with the increased number of shots. AraT5 outperforms both ChatGPT and BLOOMZ with a BLEU of 14.40.

**Question Generation and Question Answering.** For *question generation*, ChatGPT is outperformed by 0-shot performance of BLOOMZ. Nevertheless, unlike ChatGPT, BLOOMZ performance does not consistently improve as we increase the number of training shots. Compared to AraT5, both ChatGPT and BLOOMZ exhibit significantly lower scores. For *QA*, BLOOMZ significantly outperforms ChatGPT in all the few-shot settings. Specifically, BLOOMZ achieves the best score of 76.04 with 0-shot learning, whereas ChatGPT achieves 54.14 at best with 5-shot learning. However, both models are outperformed by AraT5 (81.45). We suspect BLOOMZ performs well on QA since it has been explicitly finetuned on this task using Arabic data.

**Summarization and News Title Generation.** For *summarization*, ChatGPT achieves 20.43 ROUGE, outperforming BLOOMZ by a margin of 6.87 points. Both ChatGPT and BLOOMZ are also outperformed by AraT5. For *news title generation*, ChatGPT is at 4.72 BLEU points whereas BLOOMZ struggles (1.0 BLEU). Again, AraT5 is better than both (7.72).

**Diacritization and Transliteration.** ChatGPT dominates BLOOMZ with significantly lower error rates for *diacritization*. BLOOMZ even struggles to keep the character error rate (CER) lower than

1, which means it mistakenly inserts additional characters during prediction. Although ChatGPT (CER=0.05) exhibits impressive performance on this task, AraT5 (CER=0.03) still outperforms it. For *transliteration*, ChatGPT (0.23) again outperforms BLOOMZ (0.42). However, AraT5 (0.18) achieves even lower error rates than both.

**Machine Translation.** ChatGPT outperforms BLOOMZ on all the *X* (English, Spanish, French, Russian) → *Arabic* MT tasks. As expected, both models perform better when English is used as the source language. Although the performance gap is smaller in MT as compared to the general trend in other tasks, AraT5 is still better than ChatGPT.

**Code-Switched Translation.** For Jordanian Arabic (*Jo*) mixed with English → English and Algerian Arabic (*Dz*) mixed with French → French code-switched translation tasks, ChatGPT significantly outperforms BLOOMZ by 10-30 points. ChatGPT performs slightly better for mixed English → English, while BLOOMZ performs better in mixed French → French than mixed English → English translation. Interestingly, both ChatGPT and BLOOMZ show highly superior performance than the finetuned AraT5. We suspect that both models have been pretrained with a lot of data from high-resource languages like English and French. This pretraining step helps them to perform better than the Arabic-dedicated smaller model.

**Grammatical Error Correction.** We present $M^2$Scorer $F_{0.5}$ score (Dahlmeier and Ng, 2012) for GEC in Table 2. Due to the longer sequence length ($> 4,096$), we exclude 10-shot evaluation for this task. We find ChatGPT to significantly outperform BLOOMZ (48.72 vs. 2.40 $F_{0.5}$ score). However,

AraT5 (67.54) is much better than both.

# 8 Performance on Dialectal Arabic

DA has traditionally been only spoken. It only became easier to collect dialectal data with the proliferation of social media where these varieties started to be used. Regardless, most Arabic dialects remain under-investigated due to rarity of resources. Motivated by this, we dedicate the current section to investigate performance of ChatGPT on DA as compared to MSA. In the context of this investigation, we also compare ChatGPT and GPT-4 on these two broad categories of Arabic.

For the current analysis, we first select tasks labeled as involving more DA in ORCA (i.e., all 11 tasks in Figure 3). We then run a strong in-house MSA *vs* DA classifier ($\sim 88\%$ $F_1$) to separate MSA and DA samples, keeping only samples predicted with 80% confidence. This gives us an average of 119 DA and 78.82 MSA samples across all 11 tasks. We then evaluate the macro-$F_1$ performance of ChatGPT and GPT-4 on these selected samples.
**Dialectal NLU.** To identify the extent to which GPT-4 can detect country-level dialect, we run it on our test data under 0-shot. We find GPT-4 to achieve 26.92 $F_1$, which is 13.06 improvement over ChatGPT performance reported in Table 1 (i.e, 13.86). As shown in Figure 3, both ChatGPT and

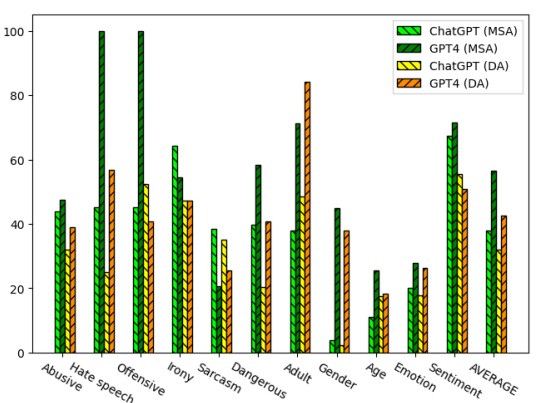

Figure 3: Comparison between ChatGPT and GPT-4 on MSA *vs* DA in macro-$F_1$ for 11 ORCA tasks.

GPT-4 perform better on MSA compared to DA (on 8 and 9 out of the 11 tasks, respectively). It is also clear that GPT-4 outperforms ChatGPT on both MSA and DA for the majority of the tasks (9 tasks for MSA and 7 tasks for DA). On average, GPT-4 improves over ChatGPT by 18.62% for MSA and 10.40% for DA tasks. *This finding suggests that (i) both* ChatGPT *and* GPT-4 *have likely seen* **more**

*MSA data than DA at some stage of their development and (ii)* GPT-4 *performs generally* **superior** *than* ChatGPT *in both MSA and DA samples.*
**Dialectal NLG.** To evaluate ChatGPT on dialectal generation, we run it on MSA and five Arabic dialects $\rightarrow$ English MT tasks from the Multi-dialectal Parallel Corpus (MDPC) proposed by Bouamor et al. (2014). Namely, we use data involving *Egyptian, Jordanian, Palestinian, Syrian,* and *Tunisian.*

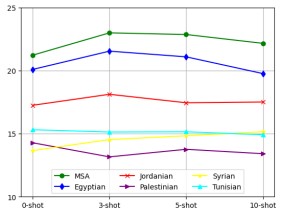

(a) *K*-shot BLEU scores of ChatGPT on MSA and 5 dialects $\rightarrow$ English MT.

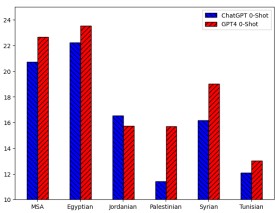

(b) Zero-shot BLEU sores of ChatGPT and GPT-4 on MSA and 5 dialects $\rightarrow$ English MT.

Figure 4: ChatGPT and GPT-4 on dialectal MT.

As Figure 4a shows, ChatGPT MSA performance is better than its dialectal performance on all our *k*-shot settings. This further substantiates what our NLU results above suggest as to ChatGPT's MSA and DA abilities. Comparing results across dialects, ChatGPT performs best on *Egyptian*, which may be directly due to availability of Egyptian dialectal data online as compared to rarity of data from other dialects (Nagoudi et al., 2022a).
**ChatGPT** *compared to* **GPT-4.** We carry out an additional set of experiments to compare ChatGPT and GPT-4 on the same five dialects and MSA test sets from Bouamor et al. (2014) listed above, but subsampling only 50 data points from each and reporting both models in zero-shot over the subsample. As Figure 4b shows, for MSA and all dialects except Jordanian, GPT-4 still outperforms ChatGPT. We also notice that GPT-4 wins with a large margin on dialects such as Palestinian, Syrian, and Tunisian, all of which are on the low-resource side compared to dialects such as Egyptian. *This finding suggests that, compared to* ChatGPT, GPT-4 *may have seen* **more** *Arabic varieties, and perhaps* **more data** *from some of the varieties.*

# 9 Human Evaluation

We also perform a set of human evaluations, motivated by the potential of human evaluation to capture subtleties of language that automated metrics may overlook. We carry out our evaluation on eight

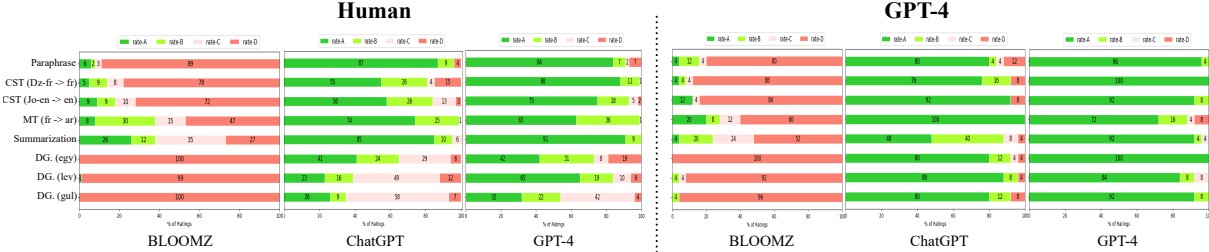

Figure 5: Evaluation of the models' responses by human and `GPT-4`. A is the best, and D is the worst rating. **MT** - Machine Translation, **CST** - Code Switched Translation, **DG** - Dialogue Generation.

NLG tasks: code-switched translation (2), dialogue generation (3), machine translation (1), paraphrase (1), and summarization (1). We particularly pick these tasks as they represent diverse generation task categories and `ChatGPT` performs poorly on some of them in our automatic evaluation setting (Table 2).

**Evaluation Setup.** We build our human evaluation framework on top of Wang et al. (2022b), Wu et al. (2023b) and implement a four-level (A, B, C, D) rating system that we adapt to language generation. (Refer to Appendix G.1 for details). We prepare the framework with examples and ask three pairs of native Arabic speakers (total=6) to rate 50 samples each from the output of each task, based on how effectively a sample has fulfilled the task described in the input prompt.

**Results.** We aggregate scores from each pair of annotators and use Cohen's kappa metric to measure the inter-annotator reliability. We find the inter-annotator agreement more than 0.6 for the majority of the tasks. We report the result in Figure 5 (Refer to Appendix L for more detailed results). From human evaluation, we find that outputs produced by `GPT-4` and `ChatGPT`, for all tasks, are rated as mostly of fine quality. We also find that `ChatGPT` is rated higher than `BLOOMZ` for summarization in the human evaluation than it is in the automated setting, perhaps reflecting that, although useful, ROUGE is not an ideal metric for summarization as it captures only token-level overlap rather than deep semantics. In addition, `ChatGPT` is rated higher than `BLOOMZ` in human evaluation than in automated evaluation for paraphrasing. This, again, reflects BLEU not being ideal for paraphrase detection for the same reasons as in the case of summarization.

**Human Analysis on CST.** As evident from Table 2, `ChatGPT` outperforms the finetuned model on the code-switched translation task. This prompted us to further probe `ChatGPT` 's code-switching abil-

ity using diagnostic test cases. Specifically, we manually evaluate `ChatGPT` 's capability of English mixed MSA, Egyptian, and Moroccan code-switched translation generation from plain English text. We ask two annotators who are fluent on the respective varieties, as well as English, to evaluate a diagnostic dataset based on *fluency* (defined as how fluent the translated text is), *faithfulness* (defined as how semantically close the translated text is to the source text), and *code-switching ability* (defined as how accurately the translated text includes code-switching). We present our result in the Table 3.

We observe that `ChatGPT` produces fluent translations that are also semantically close (faithful) to the source text. However, `ChatGPT` struggles to produce code-switched text (i.e., it generates mostly in Arabic script). Interestingly, this issue is more prevalent for MSA than Egyptian and Moroccan. We hypothesize that this discrepancy cuts across several linguistic categories and involves topics such as translation of endearment expressions; multi-word expressions, idioms, and proverbs; negation; sub-token-level code-switching; and dialectal variants of MSA lexica. We further suspect that the tokens in the source English text are very common in MSA. As a result, the model does not seem able to code-switch the words into English.

## 10 Using `GPT-4` to Evaluate Responses

Similar to Chiang et al. (2023) and Zhou et al. (2023), we assess the quality of the generated responses from `ChatGPT`, `BLOOMZ`, and `GPT-4` using the `GPT-4` model itself. For this purpose, we design a prompt (Figure 8 in Appendix E) that takes the input and the corresponding responses from the models and asks `GPT-4` to rate between 'A' and 'D' (where 'A' is the highest) for each response. (Refer to Appendix J and Appendix K for illustrative samples). We do this analysis using a random sample of

| Annotator | CST | fluency (A/B/C/D) | faithfulness (A/B/C/D) | code-switching (A/B/C/D) |
|---|---|---|---|---|
| Annotator 1 | En->MSA-En | 80/10/10/0 | 80/20/0/0 | 0/40/0/60 |
| | En->Egy-En | 70/10/20/0 | 100/0/0/0 | 20/30/0/50 |
| | En->Mor-En | 10/40/50/0 | 90/0/10/0 | 20/30/0/50 |
| | *Avg* | 53.3/20/26.7/0 | 90/6.7/3.3/0 | 13.3/33.3/0/53.4 |
| Annotator 2 | En->MSA-En | 30/70/0/0 | 90/0/10/0 | 0/30/0/70 |
| | En->Egy-En | 50/40/10/0 | 90/10/0/0 | 0/40/0/60 |
| | En->Mor-En | 20/40/40/0 | 100/0/0/0 | 20/30/0/50 |
| | *Avg* | 33.3/50/16.7 | 93.4/3.3/3.3/0 | 6.7/33.3/0/60 |
| **Average** | | 43.3/35/21.7/0 | 91.7/5/3.3/0 | 10/33.3/0/56.7 |

Table 3: Human evaluation of ChatGPT's responses on the three code-switched translation tasks.

25 data points from each of the datasets evaluated by human annotators cited in Section 9 above.[8] As Figure 5 shows, GPT-4 provides more accurate and satisfying responses (compared to ChatGPT and BLOOMZ), followed by ChatGPT, in almost all cases. BLOOMZ is rated 'D' on all of the tasks. Upon inspecting the explanations from GPT-4, we find that this poor rating of BLOOMZ is mostly due to it just copying the input samples rather than following the input instruction properly.

**Does GPT-4 eval correlate with human eval?** Chiang et al. (2023) and Zhou et al. (2023) show a positive correlation between human and GPT-4 evaluation, which we also test for Arabic NLP in this work. Hence, we compute the percentage of evaluation agreement between humans and GPT-4 on the models' responses. More specifically, for each task, we calculate at how many occurrences human evaluators agree with GPT-4 evaluation. As Figure 6 shows, at least one human evaluator agrees with GPT-4 evaluation 71.5% of the time on average. *This suggests that it may be possible to use GPT-4 to automate the time-consuming and costly process of using humans for assessing model performance. This is at least true for the tasks we consider.*

## 11 Conclusion

We presented a comprehensive evaluation of ChatGPT on 44 diverse Arabic NLP tasks. Our eval-

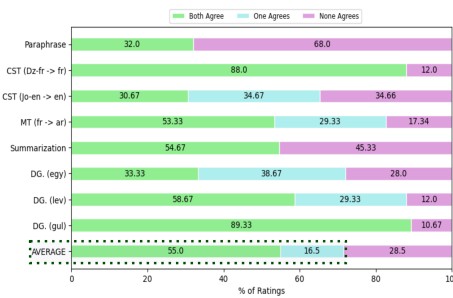

Figure 6: Human and GPT-4 agreement rate on 8 NLG tasks. On average, at least one human evaluator agrees with GPT-4 evaluation 71.5% of the time.

uation covers both NLU and NLG under $k$-shot settings. Comparing results from ChatGPT to BLOOMZ and finetuned models, we show that ChatGPT, in spite of its large size, can struggle on multiple Arabic tasks compared to much smaller finetuned models. We conduct an extensive quality assessment using both human and GPT-4 evaluation of model responses and find a positive alignment between human and GPT-4 judgment. We further investigate the performance of ChatGPT and GPT-4 on MSA *vs* DA tasks, revealing inherent limitations of both models, particularly when applied to dialects. Our findings accentuate the inherent multilingual limitations observed in both ChatGPT and GPT-4, thereby indicating a substantial scope for improving these models. We intend to expand our evaluations to encompass additional low-resource languages.

## 12 Limitations

**Experimental Setup.** In this work, we evaluate with a randomly selected subset of test samples

---

[8]This gives us 25 samples x 8 datasets = 200 decisions, for which we use the OpenAI playground. At the time of writing this paper, OpenAI allowed only 25 examples per 3 hours on GPT-4 playground.

to keep the cost manageable. Although the random selection should ideally represent the whole distribution, the performance may vary slightly when comparing to the whole test set. Additionally, ChatGPT's version can be updated on a regular interval. Hence, the results and the analyses reported here should be treated accordingly, since the model's responses can change over time (Chen et al., 2023a). To save time and cost, we perform GPT-4 generation and human, GPT-4 evaluation on a diverse but selective tasks, which we wish to extend to the other tasks in the future.

**Model Variation.** We perform evaluation on three multilingual LLMs, namely, ChatGPT, GPT-4, and BLOOMZ. Additionally, we finetune two Arabic language dedicated SOTA models i.e., MARBERT$_{V2}$ and AraT5. To keep our work resource-efficient, we do not include other massive LLMs such as PaLM 540B (Chowdhery et al., 2022). Also, since we are already comparing against the finetuned SOTA models (MARBERT$_{V2}$ and AraT5), we exclude other multilingual models like mT5 (Xue et al., 2021). However, we acknowledge that incorporating a large number of models can facilitate the comparative analysis.

**Model Evaluation.** On open-domain dialectal dialogue generation tasks, we empirically find that all the models perform poorly (close to $0.0$ BLEU) with the automatic metric. Since these tasks require the responses to be fluent and coherent and not particularly follow any closed form (e.g., MT tasks), it would not be fair to evaluate the models using such automated metrics. Therefore, we exclude the dialectal dialogue generation tasks from the automatic evaluation (Section 7) and conduct both human (Section 9) and GPT-4 evaluation (Section 10) on them.

**Findings.** Some of our findings suggest that GPT-4 can be employed to automate the process of evaluating model responses, thus replacing humans. Although this can accelerate development of AI systems and deploying them for decision making, it might also have negative implications on workforce. In particular, we recommend that our findings should be couched with caution and not be considered as a reference to replace workforce with AI.

## 13 Ethics Statement

**Data Collection and Release.** We collect the NLU evaluation datasets from ORCA. For NLG tasks, we collect from 23 publicly available datasets. To ensure proper credit assignment, we refer users to the original publications (Appendix B).

**Intended Use.** We believe our work will spur further research on studying LLMs on Arabic NLP benchmarks. As our findings show, the existing multilingual LLMs still lag behind compared to the smaller finetuned models on the majority of Arabic NLP tasks. Therefore, our work can arise the interest among the researchers to develop Arabic language dedicated LLMs that can match or outperform the SOTA finetuned models.

**Potential Misuse and Bias.** Since there exists little to no clarity on the training data of ChatGPT and GPT-4, these LLMs can produce potentially harmful and biased contents (Laskar et al., 2023). Therefore, we recommend that these models not be used in applications without careful prior consideration of potential misuse and bias.

## Acknowledgments

We acknowledge support from Canada Research Chairs (CRC), the Natural Sciences and Engineering Research Council of Canada (NSERC; RGPIN-2018-04267), the Social Sciences and Humanities Research Council of Canada (SSHRC; 435-2018-0576; 895-2020-1004; 895-2021-1008), Canadian Foundation for Innovation (CFI; 37771), Digital Research Alliance of Canada,[9] and UBC Advanced Research Computing-Sockeye.[10]

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

# A Literature Review

## A.1 Machine Translation

Language models trained on large-scale multilingual data have proven effective for a wide range of tasks spread across multiple languages. Jiao et al. (2023) evaluate ChatGPT for machine translation (MT) tasks, reporting that ChatGPT's performance on MT is on par with commercial translation tools such as google-translate. However, they also find that when translating into a distant language *pivot prompting* is very effective. In pivot prompting, instead of directly translating the source into the target, first translating the source into a high resource similar to the target language and then translating into the target is followed. Further, Jiao et al. (2023) notice that for domain-specific translation (e.g., in the biomedical filed), ChatGPT's performance degrades considerably. Peng et al. (2023) find that ChatGPT performs reasonably well in high-resource and domain-specific settings but providing additional information can be vital. Gao et al. (2023) corroborate observations of Peng et al. (2023) by designing prompts that include information such as domain, finding it to improve the MT performance of ChatGPT. Extensive evaluation by Hendy et al. (2023) shows that ChatGPT is quite good for translating into high-resource target languages but its performance degrades for low-resource languages. Zhu et al. (2023) evaluate ChatGPT and other LLMs such as XGLM (Lin et al., 2022), OPT (Zhang et al., 2022), and BLOOMZ (Muennighoff et al., 2022a) showing that even though ChatGPT is the best zero-shot and in-context few-shot model considered, it lags behind full supervised no language left behind models (NLLB) (Team et al., 2022).

## A.2 QA

ChatGPT demonstrates impressive results on question-answering tasks including user queries where no context is provided. Zheng et al. (2023) evaluate ChatGPT on complex open-domain QA tasks, studying the failures of ChatGPT and proposing methods to improve the faithfulness of its answers. Tan et al. (2023) showcase ChatGPT's limitations on the knowledge-intensive tasks which require math and reasoning skills. They show that ChatGPT is far behind the fully supervised state-of-art models on these reasoning tasks. Further evaluations of ChatGPT by Shen et al. (2023) on a wide range of QA tasks demonstrate ChatGPT's perfor-

mance across various domains. The authors show that `ChatGPT` underperforms on domain-specific QA and is also very susceptible to adversarial examples and perturbation. Omar et al. (2023) evaluate `ChatGPT` for QA on knowledge graphs. They notice that even though it falls behind the supervised SOTA methods, it can be a robust QA system for knowledge graphs.

## A.3 Text Classification

Text classification is one task where `ChatGPT` does exceptionally well in zero-shot and in-context few-shot settings. It is often even on par with full-supervised models. Zhong et al. (2023b) evaluate `ChatGPT` on GLUE NLU benchmark (Wang et al., 2019) and compare it against fully supervised BERT (Devlin et al., 2019) and RoBERTa (Liu et al., 2019) baselines. Authors conclude that `ChatGPT` outperforms BERT and RoBERTa on MNLI, SST2, and RTE while underperforming these models on other GLUE tasks. Further, `ChatGPT`'s evaluation by Gilardi et al. (2023) shows that it can outperform well-trained human annotators and crowd-workers for text classification tasks such as relevance, stance, topics, and frame detection. Wang et al. (2023b) test `ChatGPT` on a wide range of sentiment analysis datasets and find that zero-shot `ChatGPT` is on par with the fully supervised BERT model but lags behind the SOTA. The non-deterministic nature of `ChatGPT` prompted Reiss (2023) to assess the reliability and consistency of `ChatGPT` for text classification tasks in the zero-shot setting. They argue that zero-shot outputs for text classification do not meet the scientific threshold of reliability. The authors also find that minor alterations in prompts can change `ChatGPT` output. They recommend pooling the outputs from multiple repetitions to improve reliability.

Ziems et al. (2023) investigate the zero-shot ability of `ChatGPT` across a broad spectrum of computational social science benchmarks encompassing various subject areas including sociology, psychology, literature, history, linguistics, and political science. They find that `ChatGPT` demonstrates poor performance on tasks characterized by structural complexity (e.g., event arguments) or those entailing subjective expert taxonomies (e.g., hate, empathy). However, `ChatGPT` attains high performance on tasks that involve either objective ground truth (like fact-checking tasks) or explicit definition labels (e.g., anger in emotion detection). Furthermore, it is observed that `ChatGPT` exhibits a tendency to predict a neutral label that is easily recognized in the colloquial language (e.g., stereotype in the hate speech detection task), instead of utilizing a more precise label derived from the provided taxonomy (e.g., white grievance). Qin et al. (2023b) evaluate `ChatGPT` on 20 NLP datasets encompassing seven task categories in a zero-shot setting. Although `ChatGPT` performs less effectively than models fine-tuned specifically for each task, it demonstrates superior reasoning capabilities compared to other instructed finetuned models (e.g., FLAN) in natural language inference, arithmetic reasoning, and QA tasks.

**Robustness and Generalization**. `ChatGPT` does well on a wide range of downstream NLP tasks often yielding SOTA results if prompted properly. However, several studies reveal that its performance downgrades on domain-specialized tasks and is very susceptible to adversarial examples. Wang et al. (2023a) attempt to test the robustness and generalization capability of `ChatGPT`. The authors evaluate `ChatGPT` on AdvGLUE Wang et al. (2022a) and ANLI Nie et al. (2020) for adversarial robustness. They perform an evaluation on Flipkart Review and DDXPlus medical diagnosis for the out-of-distribution (OOD) generalization. The results show that `ChatGPT` outperforms all considered models (including zero-shot and fully supervised SOTA models) on every task in both settings. Chen et al. (2023b) observe on average 35.74% and 43.59% performance drop in NLI and sentiment analysis tasks, respectively, which further highlights `ChatGPT`'s vulnerability to challenging and complex scenarios.

## A.4 Multilinguality

`ChatGPT` is adept at generating high-quality responses to user queries. It also demonstrates impressive capabilities in multiple languages. Lai et al. (2023) evaluate `ChatGPT` on seven different tasks and 37 different languages belonging to low, medium, and high resource families. The results show that `ChatGPT` is either at par or even better in some tasks compared to fully supervised SOTA baselines, particularly for the high-resource languages. They also observe that for the low-resource language, providing task descriptions in a high-resource language can be helpful. The work on the multilingual evaluation of `ChatGPT` by Bang et al.

(2023) establishes that it is better at understanding non-Latin scripts but poor at generating them. Huang et al. (2023) propose cross-lingual thought prompting to improve the multilingual capabilities of ChatGPT and other LLMs. The experiments by Wu et al. (2023a) show that through careful exploitation of domain knowledge, ChatGPT can outperform the average human score on the China National Medical Licensing Examination (CNMLE). Similarly, Kasai et al. (2023) evaluate ChatGPT and other GPT family LLMs on Japanese national medical licensing examinations. They show that GPT-4 passes all six years of the exams, showcasing LLMs' impressive capability in a language that is typologically distant from English. However, they notice that ChatGPT and GPT3 fail to reach passing criteria and are prone to choosing prohibited options.

Although several works (Alammary, 2022; Abu Farha and Magdy, 2021) benchmark transformer models on Arabic understanding tasks like sentiment analysis, to the best of our knowledge, our work is the first to evaluate ChatGPT on Arabic NLU and NLG at scale.

## B  Dataset

### B.1  NLU Tasks

We evaluate on four task clusters from ORCA (El-madany et al., 2022), as follows:[11]

**Sentence Classification.**  This cluster involves the following tasks and datasets *(1) Sentiment Analysis:* (Abdul-Mageed et al., 2021b). *(2) Social Meaning:* Refers to eight social meaning datasets covering prediction of hate and offensive language detection (Mubarak et al., 2020), dangerous speech (Alshehri et al., 2020), sarcasm (Farha and Magdy, 2020), adult content (Mubarak et al., 2021), irony (Ghanem et al., 2019), emotion, age and gender (Mohammad et al., 2018; Abdul-Mageed et al., 2020b). *(3) Dialect Identification:* Involves three dialect classification levels, binary-level (i.e., MSA vs. DA), region-level (four regions), and country-level (21 countries). The three tasks are built using six datastes: ArSarcasm$_{Dia}$ (Farha and Magdy, 2020), AOC dataset (Zaidan and Callison-Burch, 2014), NADI-2020 (Abdul-Mageed et al., 2020a), MADAR (Bouamor et al., 2019), QADI (Abdelali et al., 2020), and Habibi (El-Haj,

---

[11]From ORCA, we exclude topic classification datasets since these typically have long sequences that can often exceed ChatGPT maximum length of 4,096 tokens.

---

2020). *(4) Claim Prediction*: ANS-claim (Khouja, 2020). *(5) Machine Generation,* for machine-generated text detection (Nagoudi et al., 2020).

**Paraphrase Detection.**  The goal of this cluster is to identify the similarity between a pair of sentences from a semantic perspective. This cluster contains a semantic text similarity (STS) task and a paraphrase classification task. For our evaluation, we exclude STS, which is a regression task and experiment with *paraphrase classification* using Arabic Q2Q (Seelawi et al., 2019).

**Natural Language Inference.**  This cluster covers the following two tasks: *(1) Arabic NLI:* Determining whether a text (hypothesis) is false (contradiction), undetermined (neutral), or true (entailment), given a text (premise). This task uses the Arabic part of XNLI corpus (Conneau et al., 2018). *(2) Fact-checking*: The two datasets Unified-FC (Baly et al., 2018) and ANS (Khouja, 2020) are used to target stance and factuality prediction of claims from news and social media.

**Word Sense Disambiguation (WSD).** The Arabic WSD benchmark (El-Razzaz et al., 2021), an MSA context-gloss pair dataset, is used for this task.

### B.2  NLG Tasks

For NLG, we create a benchmark using a collection of 23 publicly available datasets from different genres. We arrange our NLG datasets into 13 different task clusters:

**Machine Translation and Dialect Translation.** The MT cluster is built around the tasks of $X \rightarrow MSA$, where we test the ability of ChatGPT to translate from four foreign languages into MSA. For this, we use the United Nations Parallel Corpus (Ziemski et al., 2016) that covers the six official UN languages: Arabic, English, French, Russian, and Spanish. The dialectal translation cluster consists of *Arabic Dialects $\rightarrow$ English*, where we focus on MT from five Arabic dialects into English using the Multi-dialectal Parallel Corpus (MDPC) proposed by Bouamor et al. (2014). MDPC is a human-translated collection of 1K sentences in Egyptian, Tunisian, Jordanian, Palestinian, and Syrian Arabic, in addition to English.

**Code-Switching.**  The purpose of the code-switching (CS) task is to translate Arabic dialectal text involving code-switching from a foreign language into that foreign language. We use two human-written (natural) code-switched parallel test

sets proposed by (Nagoudi et al., 2022b): *(1) DZ-FR → FR.* It consists of code-switched Algerian dialect and French Twitter posts. These posts are manually translated into monolingual French. *(2) JO-EN → EN.* This is collected from Jordanian Twitter and consists of code-switched Jordanian dialect and English posts, which are manually translated into monolingual English.

**Summarization and Title Generation.** For this cluster, we use XLSum (Hasan et al., 2021), a diverse, multilingual summarization dataset from BBC news supporting 44 languages (including Arabic). The Arabic part of XLSum is divided into 37.5K for Train and 4.7K for each of the Dev and Test splits. The news articles in XLSum are annotated with summaries and titles, allowing us to use the articles and corresponding titles to evaluate our title generation models.

**Question Answering and Generation.** For both of these tasks, we use TyDiQA (Artetxe et al., 2020), a publicly available, multilingual, human-translated QA datasets. For the QA task, we use (*Input:* passage, question, and *Output:* answer) triplets. For the QG task, we switch these (as in *Input:* passage, answer, and *Output:* question).

**Transliteration.** The goal of transliteration (TS) is to accurately convert a word or text from one writing system to another, while maintaining the original language's pronunciation and sound. For that, we use ANETAC dataset (Ameur et al., 2019), an English-Arabic named entity transliteration dataset. It includes $79,924$ pairs of named entities in English and Arabic, which are categorized into three classes: person, location, or organization.

**Paraphrasing.** For this task, we use TaPaCo (Scherrer, 2020) a paraphrase corpus that comprises 73 languages, including Arabic. It was extracted from the Tatoeba database and created by aligning sentences with the same meaning. The Arabic portion of TaPaCo, called AraTaPaCo, contains 3K pairs of sentences.

**Text Rewriting.** The main objective here is to produce a text in the target style while maintaining the content of the original input text. We use the Arabic Parallel Gender Corpus (APGC), which was proposed by Alhafni et al. (2022). This corpus contains pairs of sentences where the input sentence is in one gender (e.g., male) and the target sentence has the same meaning but is in the opposite gender (i.e., female).

**Grammatical Error Correction.** For this task, we

use QALB 2014 (Mohit et al., 2014), a manually corrected collection of Arabic texts from online comments written by native Arabic speakers (L1) in Aljazeera articles. The dataset is divided into a training set with 19.4K sentences, a development set with 1.02K sentences, and a test set with 968 sentences.

**Dialogue Generation.** We use the open-domain dialogue generation dataset for Arabic dialects proposed by (Naous et al., 2023). The dataset consists of 1K pairs of utterances and responses, which were translated from the English DailyDialog dataset (Li et al., 2017) by three native translators from the Levantine, Egyptian, and Gulf areas.

**Diacritization.** Arabic Text Diacritization (ATD) is the process involving adding missing diacritics to words or word sequences in Arabic orthography. To accomplish this, we use the Arabic Diacritization dataset proposed by (Fadel et al., 2019).

## C    Anisotropy in BERT Models

A prevalent issue with language models is that they suffer from *anisotropy* (Ethayarajh, 2019; Li et al., 2020) in the embedding space. That is, representations obtained by the models tend to occupy a narrow cone in the hyperspace, making them less informative. This can potentially impact negatively on tasks like WSD because the models need to differentiate the representation of the queried *word* from the representation of the whole *sentence*. Forming a good representation of the queried *word* can help the model to align with the given *sense* during the finetuning. As the representation of both the *word* and the *sentence* are very close due to anisotropy, the model cannot properly align the *word* with the given *sense* even after finetuning. Therefore, we suspect that anisotropy might potentially be the reason behind the inferior performance of MARBERT$_{V2}$ on WSD task (Table 1).

## D    ChatGPT Exhibits False-Toxicity

As Table 1 shows, ChatGPT exhibits significantly poor performance compared to the finetuned AraT5 model on the toxic text classification tasks. Given the concern that models such as ChatGPT might not be aligned well for toxic and harmful language, we focus our analysis in part on the toxic language datasets in our evaluation benchmark. To this end, we compute the confusion matrices presented in Figure 7. As we can see, ChatGPT is extremely prone to flagging non-toxic texts as toxic (i.e., a

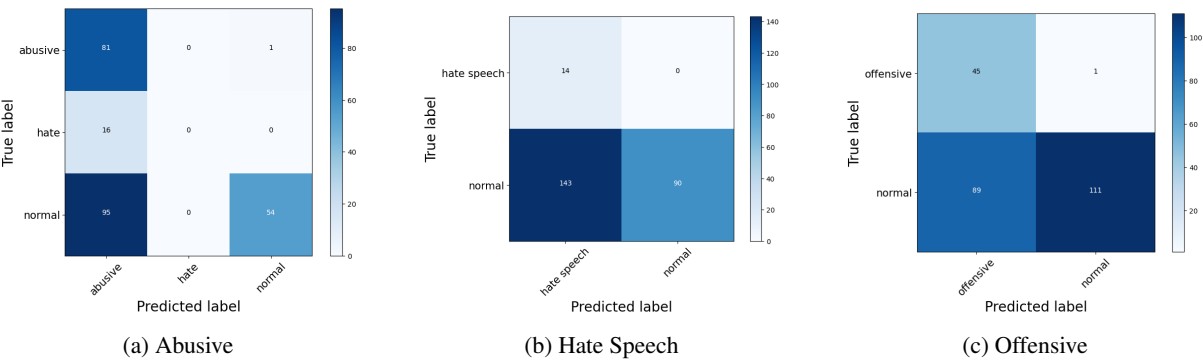

Figure 7: Confusion Matrix on the toxic text classification tasks.

high rate of false positives). We hypothesize that this may be due to one or more of the following reasons:

- Lack of *diverse* Arabic texts in `ChatGPT` pre-training data (e.g., not enough data from certain dialects).

- `ChatGPT` is supervised to alleviate the spread of harmful contents, as documented in OpenAI safety standards [12].

## E Prompt Template for `GPT-4` Evaluation

Figure 8 presents the prompt template that we use to rate the model's reponse with `GPT-4`.

## F Class Distribution for Few-shot Examples

We provide class distribution (in percentage) of the classification tasks (except for Dialect-Country where the number of classes is higher than the number of shots) for the full training dataset and our 10-shot sampling in Table 4 and Table 5, respectively.

As evident from Table 4 and Table 5, the class distribution of the few-shot samples is reasonably aligned with the corresponding tasks. Therefore, the sampling process of few-shot prompting indeed respects the class distribution of the full training set for the respective tasks in the ORCA benchmark.

## G Human Evaluation Framework

### G.1 Task Description

**Summarization:** Given a long text, we ask the model to summarize it. If we are not providing the length/size of the summary in the prompt summary

of any length shall be accepted as long as it summarizes the input.

**Paraphrasing:** Given the input text provided, we ask the model to generate a paraphrase for the input.

**Machine Translation:** Given an input in a language, we ask the model to translate it into Arabic (or any other language).

**Text Re-rewriting:** Given an input text, we ask the model to rewrite in a specific style. If we are not providing the length/size of the rewriting in the prompt, the output of any length shall be accepted as long as it is accurate without missing/adding new information on its own.

**Open-ended Dialogue Generation:** For each given utterance prompt, the task is to generate a reply. The reply is open-ended and is acceptable as long as it is coherent with the prompt.

## H Examples of Prompt

We present some sample prompts of Arabic NLU and NLG tasks that we use for `ChatGPT` evaluation in Table 6 and Table 7 respectively.

## I Examples of Models' Response

We present the examples of models' responses to the corresponding prompt in Table 8.

## J Examples of `GPT-4` Evaluation

We present the examples of `GPT-4` evaluation to the corresponding prompt in Table 9.

## K `GPT-4` Explanation on Models' Evaluation

In addition to, rate the models' responses, we generate the rationale behind the ratings by `GPT-4`. We manually analyze a subset of responses. We find

---

[12]https://openai.com/safety-standards

Figure 8: Prompt templates for GPT-4 evaluation. We generate the ratings with GPT-4 for three models (ChatGPT, BLOOMZ, and GPT-4) given the input example.

| task (full dataset) | class_0 | class_1 | class_2 | class_3 | class_4 | class_5 | class_6 | class_7 |
|---|---|---|---|---|---|---|---|---|
| dialect-binary | 59.31 | 40.69 | | | | | | |
| machine-gen | 49.87 | 50.13 | | | | | | |
| dialect-region | 31.0 | 0.07 | 44.17 | 24.76 | | | | |
| abusive | 8.02 | 29.19 | 62.79 | | | | | |
| hate speech | 5.12 | 94.88 | | | | | | |
| offensive | 79.95 | 20.05 | | | | | | |
| irony | 48.03 | 51.97 | | | | | | |
| sarcasm | 15.91 | 84.09 | | | | | | |
| dangerous | 71.16 | 28.84 | | | | | | |
| adult | 11.86 | 88.14 | | | | | | |
| gender | 53.31 | 46.69 | | | | | | |
| age | 33.98 | 30.26 | 35.76 | | | | | |
| claim | 67.5 | 32.5 | | | | | | |
| emotion | 29.62 | 1.8 | 13.75 | 15.13 | 13.3 | 2.43 | 8.44 | 15.53 |
| sentiment | 55.04 | 31.9 | 13.06 | | | | | |
| mq2q | 0.01 | 44.76 | 55.23 | | | | | |
| stance | 34.05 | 63.57 | 2.38 | | | | | |
| xnli | 33.05 | 33.69 | 33.26 | | | | | |
| wsd | 49.99 | 50.01 | | | | | | |

Table 4: Class distribution for ORCA classification tasks.

that the ratings of GPT-4 are based on generation quality, fluency, accurateness, coherency, and overall instruction-following capability. We present such examples of GPT-4's explanations on the generated ratings to the models' response in Table 10.

# L    Human Evaluation

We provide the detail human evaluation on the 8 datasets in Table 11

| task (10-shot) | class_0 | class_1 | class_2 | class_3 | class_4 | class_5 | class_6 | class_7 |
|---|---|---|---|---|---|---|---|---|
| dialect-binary | 60.0 | 40.0 | | | | | | |
| machine-gen | 50.0 | 50.0 | | | | | | |
| dialect-region | 40.0 | 0.0 | 40.0 | 20.0 | | | | |
| abusive | 10.0 | 20.0 | 70.0 | | | | | |
| hate speech | 10.0 | 90.0 | | | | | | |
| offensive | 80.0 | 20.0 | | | | | | |
| irony | 40.0 | 60.0 | | | | | | |
| sarcasm | 20.0 | 80.0 | | | | | | |
| dangerous | 70.0 | 30.0 | | | | | | |
| adult | 10.0 | 90.0 | | | | | | |
| gender | 60.0 | 40.0 | | | | | | |
| age | 40.0 | 30.0 | 40.0 | | | | | |
| claim | 80.0 | 20.0 | | | | | | |
| emotion | 50.0 | 0.0 | 10.0 | 10.0 | 20.0 | 0.0 | 10.0 | 0.0 |
| sentiment | 60.0 | 20.0 | 10.0 | | | | | |
| mq2q | 0.0 | 30.0 | 70.0 | | | | | |
| stance | 40.0 | 50.0 | 10.0 | | | | | |
| xnli | 40.0 | 30.0 | 30.0 | | | | | |
| wsd | 50.0 | 50.0 | | | | | | |

Table 5: Class distribution for 10-shot sampling.

| Rating | Criteria |
|--------|----------|
| Rating-A | • The output is an acceptable, valid, and satisfying response to the input prompt. Eg: If we ask the model to summarize an input, the produced output should clearly summarize the input text. For paraphrasing and machine translation, the output is fluent and contains the same semantic meaning as the input.
• Output is fluent, relevant, and natural response to a conversation.
• The produced output is correct but at the same standard as humans. This is applicable for tasks like open-ended dialogue generation. Irrespective of language and tone (layman term vs domain expert, formal or informal), as long as the generated response meets criteria 1 it should be rated as A. |
| Rating-B | • It has followed the task given in the prompt but has a minor issue that needs to be improved.
• Partially acceptable responses shall be labeled as B.
• output has fulfilled the task specified in the prompt with >= 50% accuracy.
• The task-specific criteria for 'B' is as follows:
**Summarization:** The model has omitted some key information from the input text in the summary. The model has added some information in the summary that's not in the input text. Minor overlap between summary and text input.
**Paraphrasing:** The output has minor grammar issues. Minor overlap between input text and generated paraphrase.
**Open-ended Dialogue Generation:** The generated response has minor factual or syntactical issues. The generated response has little irrelevant information.
**Machine Translation:** The output has minor fluency issues. The output has minor syntactical issues. The output may contain few irrelevant information. |
| Rating-C | • The output is relevant and the model attempts to do the task specified in the prompt but has a major issue in the quality of the output.
• The produced output is <50% accurate for the task specified in the prompt.
• The task-specific criteria for 'C' is as follows:
**Summarization:** The model has omitted significant information from the input text in the summary. The model has added significant information in the summary that's not in the input text. Major overlap between summary and text input.
**Paraphrasing:** The output has major grammar issues. The output is the same as input text with a few minor word replacements. Major overlap between input text and generated paraphrase.
The output has a major semantic meaning difference compared to the input.
**Open-ended Dialogue Generation:** The generated response has major factual or syntactical issues. Generated response is machine-like, non-fluent, and/or irrelevant.
**Machine Translation:** The produced translation may contain one or two different language tokens except for the target language. The output has major grammatical, syntactical, and fluency issues. The semantic meaning is a little different than the input. |
| Rating-D | • The output is invalid and totally unacceptable for the task specified in the prompt.
• The produced output is not at all relevant to the task specified in the prompt. Eg: The model didn't produce any output at all.
• The model has refused to provide an answer. Eg: "As an AI language model I do not have sufficient ..."
• The model just copies the input as the output
• The produced output is in a different language. |

| Task | Shot | Prompt |
|---|---|---|
| Abusive | 0 | You are an annotatorGPT. Your task is to do abusive language classification. You will be given an input text and you should predict the label of the text from the list: ['hate', 'normal', 'abusive']. 
 Now predict the label of the following input: 
 Input:شعوب بتسوا صرامي 
 Output: |
| Dialect Binary | 0 | You are an annotatorGPT. Your task is to do dialect classification between Dialectal Arabic (DA) vs Modern Standard Arabic (MSA). You will be given an 
 input text and you should predict the label of the text from the list: ['DA', 'MSA']. 
 Now predict the label of the following input: 
 Input:الف مبروك للنادي بني حسن عشيره 
 Output: |
| Paraphrase | 0 | You are an annotatorGPT. Your task is to do duplicate/paraphrase detection. You will be given two input texts and you should predict the label from the list: ['duplicates', 'not duplicates']. Now predict the label of the following input: 
 Input: 
 Text 1: كم يبلغ طول سور الصين العظيم ؟ 
 Text 2: ما هي الإمتدادات الحدزدية لسور الصين العظيم ؟ 
 Output: |
| Sarcasm | 0 | You are an annotatorGPT. Your task is to do sarcasm text classification. You will be given an input text and you should predict the label of the text from the list: ['sarcastic', 'non-sarcastic']. 
 Now predict the label of the following input: 
 Input: هناك مكان اسفل الجحيم للي بيتفرجوا على فيفي عبده 
 Output: |
| Sentiment | 3 | You are an annotatorGPT. Your task is to do sentiment classification. You will be given an input text and you should predict the label of the text from the list: ['positive', 'negative', 'neutral']. Here, we provide you 3 examples of the task: 
 Example: 1 
 Input:راجعنا بعد سنتين 
 Output: negative 
 Example: 2 
 Input: في جولة ميدانية لسبق.. سائقون يبحثون عن الإطار الأرخص 
 Output: neutral 
 Example: 3 
 Input: USER ماشاء الله كيف امداهم يسوون هالعمل 
 البطولي بهالفترة شكرا امانة الرياض مره مره كلفتوا على نفسكم 
 Output: negative 
 Now predict the label of the following input: 
 Input: وأخير والحمد لله نجحتوا 
 Output: |

Table 6: Prompt examples for some Arabic NLU tasks.

| Task | Shot | Prompt |
|---|---|---|
| Question Gen | 0 | You are a generative language model. Your task is to do question generation. You will be given a context and the answer of the question as input. You should generate the question based on the context and the answer. The output must be in Arabic. You do not need to provide any explanation.
Now generate the output of the following input:
Input:
Context: ولدت عام ١٧٥٥ في فيينا، ثم انتقلت إلى فرنسا لتتزوج،
وهي أصغر أبناء الملكة ماريا تيريزا ملكة . تزوجت ماري أنطوانيت من
الملك لويس السادس عشر وهي في الرابعة عشر من عمرها وكان هو في
الخامسة عشرة من عمره.
Answer:فيينا
Output: |
| Diacritization | 0 | You are a generative language model. Your task is to do text diacritization. You will be given an input Arabic text and you should generate the diacritized version of the text as output. The output must be in Arabic. You do not need to provide any explanation. Now generate the output of the following input:
Input: ( قوله لأنه حق العبد ) أي بدلالة جواز الفعل والاعتياض .
Output: |
| MT (en→ar) | 0 | You are a generative language model. You task is to do machine translation. You are given an English text as the input, you should translate it to Arabic. You do not need to provide any explanation. Now generate the output of the following input:
Input: It publishes a handbook offering a selection of first-class academic programmes and providing key information to help understand the French academic system.
Output: |
| CST (Jo-en→en) | 0 | You are a generative language model. You task is to do code switched machine translation. You are given a Jordan Arabic and English mixed text as the input, you should translate it to English. You do not need to provide any explanation. Now generate the output of the following input:
Input: القصة وما فيها : صررلاند يـهرسلف وته پستف پـهپل وبس
Output: |
| Dialogue (EGY) | 3 | You are a generative language model. You task is to do dialogue generation. You are given an input text, you should generate an output in reply to that input text. The output must be in the same Arabic dialect as input. You do not need to provide any explanation. Here, we provide you 3 examples of the task:
Example: 1
Input: حتسافر امتى؟
Output: بعد كام شهر عشان اكون خلصت جامعة
Example: 2
Input: باين على زوجك انو خدوم جدا
Output: لا متعشميش اوي
Example: 3
Input: بتحب الأكشن
Output: اه خاصة افلام العربيات
Now generate the output of the following input:
Input: راسي مش متحمل التوتر
Output: |

Table 7: Prompt examples for some Arabic NLG tasks.

| Prompt | Gold | BLOOMZ | ChatGPT | GPT-4 |
|---|---|---|---|---|
| You are a generative language model. You task is to do machine translation. You are given a French text as the input, you should translate it to Arabic. You do not need to provide any explanation. Now generate the output of the following input:
Input: annexe ii
Output: | المرفق الثاني | المرفق الثاني | الملحق الثاني | الملحق الثاني |
| You are a generative language model. You task is to do dialogue generation. You are given an input text, you should generate an output in reply to that input text. The output must be in the same Arabic dialect as input. You do not need to provide any explanation.
Now generate the output of the following input:
Input: صباح الخير يا أحمد.
Output: | هلا مريم. | صباح الخير يا أحمد. | صباح النور يا صديقي، كيف حالك اليوم؟ | صباح النور والسرور، كيفك اليوم؟ |
| You are a generative language model. You task is to do dialogue generation. You are given an input text, you should generate an output in reply to that input text. The output must be in the same Arabic dialect as input. You do not need to provide any explanation.
Now generate the output of the following input:
Input: مين هيدي المرا ؟
Output: | هيدي ستي أم ابوي | مين هيدي المرا ؟ | ما بعرف، ممكن توضح أكتر؟ | ما بعرف، ما شفتها من قبل. |
| You are a generative language model. You task is to do code switched machine translation. You are given a Jordan Arabic and English mixed text as the input, you should translate it to English. You do not need to provide any explanation.
Now generate the output of the following input:
Input: الاهل والقرايب مصيبة اكتر من ال fake acc
Output: | Family and relatives are more calamity than fake acc | الاهل والقرايب مصيبة اكتر من ال fake acc | Family and relatives are a bigger problem than fake accounts. | Family and relatives are more trouble than fake accounts. |
| You are a generative language model. You task is to do code switched machine translation. You are given a Algerian Arabic and French mixed text as the input, you should translate it to French. You do not need to provide any explanation.
Now generate the output of the following input:
Input: Bonjour أحبائي
Output: | Bonjour les amies | Bonjour mes amis | Bonjour mes chers. | Bonjour mes bien-aimés |

Table 8: Examples of models' responses.

| Prompt | GPT-4 Rating |
|---|---|
| You are a helpful and precise assistant for checking the quality of the response. {Prompt} You are a generative language model. You task is to do machine translation. You are given a French text as the input, you should translate it to Arabic. You do not need to provide any explanation. Now generate the output of the following input: Input: l'eiil a continué d'exécuter des civils. Output: {The Start of Assistant 1's Answer} واصل تنظيم الدولة الإسلامية إعدام المدنيين. {The End of Assistant 1's Answer} {The Start of Assistant 2's Answer} استمر في تنفيذ المدنيين. {The End of Assistant 2's Answer} {The Start of Assistant 3's Answer} استمر تنظيم داش في إعدام المدنيين. {The End of Assistant 3's Answer} {System} We would like to request your feedback on the performance of three AI assistants in response to the prompt displayed above. Please rate the responses based on their correctness, instruction-following capability, relevance, coherency, and fluency. Each assistant receives an overall rating on a scale of 'A' to 'D', where 'A' indicates the best overall performance and 'D' indicates the worst overall performance. For example, if the AI assistant just copy/paste the input text, you should rate it as 'D'. Please first output a single line containing only three values indicating the scores for Assistant 1, 2, and 3 respectively. The three scores are separated by a space. In the subsequent line, please provide a comprehensive explanation of your evaluation, avoiding any potential bias and ensuring that the order in which the responses were presented does not affect your judgment. | A D A |
| You are a helpful and precise assistant for checking the quality of the response. {Prompt} You are a generative language model. You task is to do code switched machine translation. You are given a Jordan Arabic and English mixed text as the input, you should translate it to English. You do not need to provide any explanation. Now generate the output of the following input: Input: Yess and my تسليك skills are getting much better now! Output: {The Start of Assistant 1's Answer} yes and my plumbing skills are getting much better now! {The End of Assistant 1's Answer} {The Start of Assistant 2's Answer} yess and my english skills are getting much better now! {The End of Assistant 2's Answer} {The Start of Assistant 3's Answer} yess and my threading skills are getting much better now! {The End of Assistant 3's Answer} {System} We would like to request your feedback on the performance of three AI assistants in response to the prompt displayed above. Please rate the responses based on their correctness, instruction-following capability, relevance, coherency, and fluency. Each assistant receives an overall rating on a scale of 'A' to 'D', where 'A' indicates the best overall performance and 'D' indicates the worst overall performance. For example, if the AI assistant just copy/paste the input text, you should rate it as 'D'. Please first output a single line containing only three values indicating the scores for Assistant 1, 2, and 3 respectively. The three scores are separated by a space. In the subsequent line, please provide a comprehensive explanation of your evaluation, avoiding any potential bias and ensuring that the order in which the responses were presented does not affect your judgment. | A D B |

Table 9: Examples of GPT-4 ratings on the models' responses.

| Task | Input | Model Response | GPT-4 Rating | GPT-4 Explanation |
|------|-------|----------------|--------------|-------------------|
| Paraphrasing | لديك أخين أصغر منك. | لديك أخ أصغر منك. | C | Assistant 2 changed the number of brothers in the paraphrase "لديك أخ أصغر منك" (You have a younger brother), which does not preserve the original semantic meaning of the input text. Hence, Assistant 2 receives a 'C' rating. It did attempt to paraphrase, but failed to retain key information from the original. |
| MT (en→ar) | autres dépenses de personnel | نومصروفاتو ظفين الأخرى | D | Assistant 3's translation, on the other hand, seems to be misspelled and grammatically incorrect. It seems that there is a typing error in the response, which causes the sentence to not make sense in Arabic. Hence, Assistant 3 is given the grade 'D'. |
| CST (Dz-fr→fr) | Il faut تتزوجي | il faut se marier | C | Assistant 2, however, incorrectly translated the Arabic text "تتزوجي" (which means "you should marry" in feminine form) into "se marier" (to marry in general) in French. This translation is not accurate because it changes the meaning and the recipient of the original sentence. Therefore, Assistant 2 gets the grade 'C'. |
| Dialogue (EGY) | go? تعرف تلعب | ايوا، بس مش بشكل كبير. | B | Assistant 1's response is brief and a little ambiguous, but it's relevant and maintains the same dialect. While it technically answers the question, it lacks some depth and detail which would have been helpful. Therefore, I give it a 'B' rating. |
| CST (Jo-en→en) | اشتقت تغنيلي lego house | I miss you singing "lego house" to me. | A | Assistant 1 correctly translates the Arabic part of the input text into English, maintaining its original context and meaning. The response is coherent, relevant, and follows the given instructions correctly, earning Assistant 1 an 'A' grade. |

Table 10: Examples of GPT-4's explanation on the evaluation of the generated responses.

| Task | Annotator | ChatGPT | | | | BLOOMZ | | | | GPT-4 | | | |
|---|---|---|---|---|---|---|---|---|---|---|---|---|---|
| | | A | B | C | D | A | B | C | D | A | B | C | D |
| Paraphrasing | 1 | 42 | 6 | 0 | 2 | 2 | 2 | 1 | 45 | 43 | 0 | 0 | 7 |
| | 2 | 45 | 3 | 0 | 2 | 4 | 0 | 2 | 44 | 41 | 7 | 2 | 0 |
| | Avg | 43.5 | 4.5 | 0 | 2 | 3 | 1 | 1.5 | 44.5 | 42 | 3.5 | 1 | 43.5 |
| | Sum | 87 | 9 | 0 | 4 | 6 | 2 | 3 | 89 | 84 | 7 | 2 | 7 |
| Summarization | 1 | 43 | 4 | 4 | 0 | 22 | 12 | 5 | 11 | 43 | 7 | 0 | 0 |
| | 2 | 42 | 6 | 2 | 0 | 4 | 0 | 30 | 16 | 48 | 2 | 0 | 0 |
| | Avg | 42.5 | 5 | 3 | 0 | 13 | 6 | 17.5 | 13.5 | 45.5 | 4.5 | 0 | 0 |
| | Sum | 85 | 10 | 6 | 0 | 26 | 12 | 35 | 27 | 91 | 9 | 0 | 0 |
| CST (Dz-fr→fr) | 1 | 26 | 14 | 1 | 9 | 2 | 5 | 4 | 39 | 43 | 6 | 1 | 0 |
| | 2 | 29 | 12 | 3 | 6 | 3 | 4 | 4 | 39 | 45 | 5 | 0 | 0 |
| | Avg | 27.5 | 13 | 2 | 7.5 | 2.5 | 4.5 | 4 | 39 | 44 | 5.5 | 0.5 | 0 |
| | Sum | 55 | 26 | 4 | 15 | 5 | 9 | 8 | 78 | 88 | 11 | 1 | 0 |
| CST (Jo-en→en) | 1 | 27 | 9 | 11 | 3 | 3 | 3 | 5 | 39 | 38 | 6 | 4 | 2 |
| | 2 | 31 | 17 | 2 | 0 | 6 | 6 | 5 | 33 | 37 | 12 | 1 | 0 |
| | Avg | 29 | 13 | 6.5 | 1.5 | 4.5 | 4.5 | 5 | 36 | 37.5 | 9 | 2.5 | 1 |
| | Sum | 58 | 26 | 13 | 3 | 9 | 9 | 10 | 72 | 75 | 18 | 5 | 2 |
| MT (Fr→Ar) | 1 | 36 | 14 | 0 | 0 | 4 | 15 | 8 | 23 | 22 | 28 | 0 | 0 |
| | 2 | 38 | 11 | 1 | 0 | 4 | 15 | 7 | 24 | 41 | 8 | 1 | 0 |
| | Avg | 37 | 12.5 | 0.5 | 0 | 4 | 15 | 7.5 | 23.5 | 31.5 | 18 | 0.5 | 0 |
| | Sum | 74 | 25 | 1 | 0 | 8 | 30 | 15 | 47 | 63 | 36 | 1 | 0 |
| Dialogue (EGY) | 1 | 26 | 17 | 6 | 1 | 0 | 0 | 0 | 50 | 23 | 14 | 4 | 9 |
| | 2 | 15 | 7 | 23 | 5 | 0 | 0 | 0 | 50 | 19 | 17 | 4 | 10 |
| | Avg | 20.5 | 12 | 14.5 | 3 | 0 | 0 | 0 | 50 | 21 | 15.5 | 4 | 9.5 |
| | Sum | 41 | 24 | 29 | 6 | 0 | 0 | 0 | 100 | 42 | 31 | 8 | 19 |
| Dialogue (LEV) | 1 | 8 | 9 | 30 | 3 | 1 | 0 | 0 | 49 | 25 | 16 | 8 | 1 |
| | 2 | 15 | 7 | 19 | 9 | 0 | 0 | 0 | 50 | 40 | 3 | 2 | 5 |
| | Avg | 11.5 | 8 | 24.5 | 6 | 0.5 | 0 | 0 | 49.5 | 32.5 | 9.5 | 5 | 3 |
| | Sum | 23 | 16 | 49 | 12 | 1 | 0 | 0 | 99 | 65 | 19 | 10 | 6 |
| Dialogue (GUL) | 1 | 18 | 4 | 27 | 1 | 0 | 0 | 0 | 50 | 25 | 9 | 14 | 2 |
| | 2 | 8 | 5 | 31 | 6 | 0 | 0 | 0 | 50 | 7 | 13 | 28 | 2 |
| | Avg | 13 | 4.5 | 29 | 3.5 | 0 | 0 | 0 | 50 | 16 | 11 | 21 | 2 |
| | Sum | 26 | 9 | 58 | 7 | 0 | 0 | 0 | 100 | 32 | 22 | 42 | 4 |

Table 11: Human evaluation results.