# OpenReview forum: "GPTAraEval: A Comprehensive Evaluation of ChatGPT on Arabic NLP"
_EMNLP/2023/Conference — EMNLP 2023 Main_

### Official Review · Reviewer_Utir · 2023-08-02

**Soundness:** 4

**Excitement:**

3: Ambivalent: It has merits (e.g., it reports state-of-the-art results, the idea is nice), but there are key weaknesses (e.g., it describes incremental work), and it can significantly benefit from another round of revision. However, I won't object to accepting it if my co-reviewers champion it.

**Paper Topic And Main Contributions:**

This work evaluates ChatGPT, BloomZ, and GPT-4 on a broad suite of NLU and NLG tasks in Arabic, covering both Dialectal and Modern Standard Arabic. It shows that these LLMs lag behind their finetuned counterparts consistently across both NLG and NLU tasks, but that GPT-4 is still a clear leader in these tasks, while open-source LLM BloomZ consistently struggles. They use both automated benchmarks and a small set of human evaluations for a subset of NLG tasks.

**Questions For The Authors:**

A) In Section 4, you mention that an array of prompt templates was evaluated, but English prompts were found to be best. How were candidate prompts evaluated? With what dataset? On all models?

B) Do you constrain the outputs of ChatGPT or BloomZ using logit bias or using logprobs rather than free-form generation? If not, how are responses matched to the label set from a free-form generation?

C) You mention that your annotators are native Arabic speakers, but don't mention their familiarity with dialects. Which dialects are they familiar with and what is their level of familiarity for each dialect? Are they familiar with the non-Arabic languages in the codeswitched evaluations and if so what is their level of familiarity with them?

D) How was inter-annotator reliability measured for each task evaluated with humans?

E) How were the finetuned models trained? Hyperparameters and training data for those models should be reported to be able to reproduce the finetuned baselines used in this work and to get a sense for how reasonable their training process was.

G) Why was AraT5 omitted from the human evaluation? This seems a significant omission since it outperforms the other models by the automatic baselines.

**Reasons To Accept:**

- The work evaluates large-scale commercial LLMs on existing NLP datasets to a new, widely spoken language. This gives future researchers baseline results that they can compare their methods to for a broad range of Arabic NLP tasks, while saving them the cost of querying these relatively expensive APIs.

- The work evaluates across both open-source models and leading closed-source APIs for Arabic NLP. This allows more practically oriented researchers to have a sense of what the tradeoffs might be when looking at different tools for Arabic NLP.

**Reasons To Reject:**

- The authors utilize a static prompt template for all tasks - but LLM prompting can result in extremely high variance and usually needs to be tuned between tasks. Without evaluating multiple prompts per task, these results may be misleading as readers have no sense what the variance is. This only compounds with the reproducibility challenges inherent to evaluations based on changing API endpoints acknowledged in the limitations. Especially since the authors draw conclusions primarily by the relative rankings of models, this work seems to have reasonable risk of some conclusions being misleading/incorrect given different prompt phrasing or as the API changes.

- The work's analysis focuses on high level comparison of the performance metrics and postulations at their causes. This means that the primary takeaway from the work is that these models have significant room to grow for Arabic NLP, but they don't offer a deeper argument for how these findings should change existing research directions or what is unique about the challenges for Arabic NLP for LLMs vs. other languages.

**Reproducibility:**

3: Could reproduce the results with some difficulty. The settings of parameters are underspecified or subjectively determined; the training/evaluation data are not widely available.

**Reviewer Confidence:**

5: Positive that my evaluation is correct. I read the paper very carefully and I am very familiar with related work.

**Typos Grammar Style And Presentation Improvements:**

The text in the figures is too small to read without significant zooming, even on a large monitor. These font sizes should be increased for readability.

Figures 3, 5 and 6 are not color-blind friendly color-schemes. This will make it for many readers to distinguish the categories. You can either use other color-schemes (and test them with browser extensions which simulate colorblindness) or add additional ways to delineate categories beyond the color itself.

Line 239 says "deterministic and reproducible results", but this is contradicted by the acknowledgement in the limitations that the API changes frequently. Furthermore, even at temperature 0.0, both ChatGPT and GPT-4 have been shown to not be deterministic. This line should be removed entirely. Ideally, it should be replaced by a more up-front acknowledgement about the reproducibility challenges of evaluating API-only access models rather than leaving this for the limitations.

---

> ### Author Rebuttal · Authors · 2023-08-29
>
> Dear Reviewer,
>
> Thank you for your thorough review. We are encouraged to know you see our paper as helping more practically-oriented researchers working on Arabic NLP. We provide detailed answers to your comments next.
>
> **W1. The authors utilize a static prompt template for all tasks - but LLM prompting can result in extremely high variance and usually needs to be tuned between tasks. Without evaluating multiple prompts per task, these results may be misleading as readers have no sense what the variance is. This only compounds with the reproducibility challenges inherent to evaluations based on changing API endpoints acknowledged in the limitations. Especially since the authors draw conclusions primarily by the relative rankings of models, this work seems to have reasonable risk of some conclusions being misleading/incorrect given different prompt phrasing or as the API changes.**
>
> In our work, we hypothesize that a good model should respond successfully to a standardized and informative prompt. We also believe that a fixed prompt, so long as it is sufficiently clear, is necessary for a fair comparison between the different models. Without this type of fixed prompt template, it is difficult to evaluate and rank different models across different tasks. For these reasons, we provided the models with a ***standard*** prompt that has also been employed previously for ChatGPT benchmarking on English tasks [1] (Table 27) and multilingual tasks [2].  Our prompt is also ***informative***: we designed a well-articulated prompt defining the roles, the tasks, and the associated expected responses for each specific task.
>
> In addition, while we agree with the reviewer that variation in prompt can at times change the performance, we do not see the pattern in our result changing in any significant way, especially given the large margin of difference (on average 17.57% for NLU and 4.46% for NLG) between the larger models and the smaller finetuned models.
>
> Regarding your point to the challenges of reproducibility associated with API changes, we believe this is external to our work. However, we have stated the exact API version we deployed (line 238) to facilitate future comparisons to the extent this is possible. Please also note that, similar to other works on ChatGPT evaluation [1][2], we do not claim our benchmarking to be universal across all variations of experimental settings. Rather, we emphasize that our evaluation and benchmarking are reproducible by using the detailed standard prompt we opt for and the exact version of the API that we employ. The impact of the variation of prompting and API versions can be considered as an incremental line of future research.
>
>
>
> **W2. The work's analysis focuses on high level comparison of the performance metrics and postulations at their causes. This means that the primary takeaway from the work is that these models have significant room to grow for Arabic NLP, but they don't offer a deeper argument for how these findings should change existing research directions or what is unique about the challenges for Arabic NLP for LLMs vs. other languages.**
>
> This is useful feedback and we will include a dedicated section focusing on offering a deeper argument for how our findings can change existing research directions. The reviewer is also right that in our work, (1) we show a significant room for improvement for LLMs like ChatGPT on Arabic (lines 114-119).
>
> Meanwhile, our work offers a number of other important takeaways that can guide future research. For example, since Arabic is a collection of languages and language varieties, (2) we clearly show that LLMs do not handle tasks involving the different varieties and dialects of Arabic equivalently. For this reason, we suggest that future research should focus more on Arabic dialects. This includes involving dialectal pretraining data in these models as well as possibly finetuning models on dialectal tasks (lines 106-109; lines 487-489; lines 501-505). We also delve deeper into this analysis, offering a (3) dedicated section showing the unique challenge associated with Arabic NLP based on the fact that the performance differs between modern standard Arabic and dialectal Arabic (Section 8). To the best of our knowledge, this is the first work to study the comparison of ChatGPT and GPT-4 on Arabic dialectal variation (lines 82-85). (4) We also emphasize the role of GPT-4 as a potential alternative of human evaluators for Arabic language (Section 10).
>
> **Questions For The Authors:**
>
> **QA. In Section 4, you mention that an array of prompt templates was evaluated, but English prompts were found to be best. How were candidate prompts evaluated? With what dataset? On all models?**
>
> We thank the reviewer for this question. We evaluated Arabic and English prompt templates on both NLU and NLG tasks with ChatGPT and BLOOMZ. Specifically, we evaluated on dialect identification, machine-generated text detection, and toxicity detection for NLU and machine translation tasks for NLG. We evaluate the models on these tasks with both English and Arabic prompts, observing that the English prompt outperforms its Arabic counterpart. This finding is also substantiated by prior literature [2] that observed the superiority of the English prompt even on multilingual tasks (lines 146-150). We will add this explanation in the camera-ready version.
>
> **QB.  Do you constrain the outputs of ChatGPT or BloomZ using logit bias or using logprobs rather than free-form generation? If not, how are responses matched to the label set from a free-form generation?**
>
> We thank the reviewer for raising this point. We did not use logit bias or logprobs. Rather, we used free-form generation for both ChatGPT and BloomZ. This is because constraining the models to specific output would undermine the instruction-following capability of the models. Hence, we follow [3] to directly predict the answer with free-form generation along with simple post-processing like removing leading or trailing whitespace, and then apply exact matching with the label set. We will add this response in the experimental setting of our paper.
>
> **QC. You mention that your annotators are native Arabic speakers, but don't mention their familiarity with dialects. Which dialects are they familiar with and what is their level of familiarity for each dialect? Are they familiar with the non-Arabic languages in the code-switched evaluations and if so what is their level of familiarity with them?**
>
> The annotators who contributed to this work are native speakers of the respective dialects they evaluated. The native speakers who evaluated Algerian-French to French translation are native Algerian speakers who have a professional level of expertise in French. The same applies to other code-switching tasks.
>
> **QD. How was inter-annotator reliability measured for each task evaluated with humans?**
>
> We thank the reviewer for this question. We used Cohen’s kappa metric to measure the inter-annotator reliability. We find the inter-annotator agreement more than 0.6 for the majority of the tasks. We will add the details of inter-annotator agreement to Section 9 of the paper.
>
> **QE. How were the finetuned models trained? Hyperparameters and training data for those models should be reported to be able to reproduce the finetuned baselines used in this work and to get a sense for how reasonable their training process was.**
>
> We apologize for not including these important details about models’ finetuning. For both MARBERT and AraT5, we identified the best model on the respective development split (Dev) and blind-test on the testing split (Test). We train MARBERT for 25 epochs with a patience of 5 and AraT5 for 10 epochs. For both models, we set the learning rate of 5e-5 across all the tasks. We will add this information in Section 5 to facilitate reproducibility.
>
> **QG. Why was AraT5 omitted from the human evaluation? This seems a significant omission since it outperforms the other models by the automatic baselines.**
>
> This is a great question! In our automated evaluation, we find that the finetuned AraT5 significantly outperforms ChatGPT (Table 2). Hence, we pick a number of tasks that are both (1) diverse and (2) cause ChatGPT to perform poorly for human evaluation (lines 528-532). Since our focus is to understand the capability of large language models on language generation in zero-shot settings, we exclude AraT5 from the human evaluation. More concretely, we focus on the LLMs that perform poorly in order to investigate whether the reason originally stems from their lack of generalizability on unseen data or the unavailability of proper automated metrics to measure their quality.
>
> **Typos Grammar Style And Presentation Improvements:**
>
> **T1. The text in the figures is too small to read without significant zooming, even on a large monitor. These font sizes should be increased for readability.**
>
> We apologize for the inconvenience. We will improve the font size in the figures.
>
> **T2. Figures 3, 5 and 6 are not color-blind friendly color-schemes. This will make it for many readers to distinguish the categories. You can either use other color-schemes (and test them with browser extensions which simulate colorblindness) or add additional ways to delineate categories beyond the color itself.**
>
> Thank you for the suggestion. We will update the mentioned figures with more color-blind-friendly schemes.
>
> **T3. Line 239 says "deterministic and reproducible results", but this is contradicted by the acknowledgement in the limitations that the API changes frequently. Furthermore, even at temperature 0.0, both ChatGPT and GPT-4 have been shown to not be deterministic. This line should be removed entirely. Ideally, it should be replaced by a more up-front acknowledgement about the reproducibility challenges of evaluating API-only access models rather than leaving this for the limitations.**
>
> We thank the reviewer for the suggestion. We mentioned the API version (line 238) that we used in our work. However, we will add an acknowledgment outlining the challenge of reproducibility of the closed-models.
>
> **References**
>
> [1] Laskar et al. A Systematic Study and Comprehensive Evaluation of ChatGPT on Benchmark Datasets. ACL 2023.
>
> [2] Lai et al. Chatgpt beyond English: Towards a comprehensive evaluation of large language models in multilingual learning. arXiv 2023.
>
> [3] Chung et al. Scaling Instruction-Finetuned Language Models. arXiv 2022.

---

### Official Review · Reviewer_YeJa · 2023-08-04

**Typos Grammar Style And Presentation Improvements:** NLI is used in line 160 for the first…
**Soundness:** 4

**Excitement:**

4: Strong: This paper deepens the understanding of some phenomenon or lowers the barriers to an existing research direction.

**Paper Topic And Main Contributions:**

The paper presents a set of experiments evaluating the performance of ChatGPT on varying Arabic NLP tasks, in both Standard and Dialectal Arabic. The experiments are run on ChatGPT and GPT-4, as well as on other LLMs. The paper also provides a human evaluation to compare the results of varying systems.

**Reasons To Accept:**

Understanding how ChatGPT is performing on varying languages is crucial in understanding how to improve LLMs output on new languages. The paper presents a large number of experiments for various NLP tasks that cover a wide range of real-world problems.

**Reasons To Reject:**

Reading through the description of the varying tasks and their results was a bit tedious since there are so many tasks to cover.

**Reproducibility:**

3: Could reproduce the results with some difficulty. The settings of parameters are underspecified or subjectively determined; the training/evaluation data are not widely available.

**Reviewer Confidence:**

4: Quite sure. I tried to check the important points carefully. It's unlikely, though conceivable, that I missed something that should affect my ratings.

---

> ### Author Rebuttal · Authors · 2023-08-29
>
> Dear Reviewer,
>
> Thank you for your helpful comments. We are pleased to know that you liked the thorough experimental evaluation of this work. Please find our response to your concerns below:
>
>
> **W1. Reading through the description of the varying tasks and their results was a bit tedious since there are so many tasks to cover.**
>
> Thank you for your comment. Our intention is to offer a comprehensive evaluation on a broad range of Arabic NLP tasks. We needed to include sufficient description of various tasks and the results we acquired. We believe this is necessary for replication and facilitation of meaningful comparisons that may be conducted in the future. However, we had actually put a very concise description of the tasks in Section 3 and moved the detailed description in Appendix B.
>
> **Typos Grammar Style And Presentation Improvements**
>
> *T1. NLI is used in line 160 for the first time without expanding the acronym.*
>
> We thank the reviewer for pointing this out. We will spell out “natural language inference” before we introduce the acronym.

---

### Official Review · Reviewer_ikv4 · 2023-08-05

**Soundness:** 4

**Excitement:**

3: Ambivalent: It has merits (e.g., it reports state-of-the-art results, the idea is nice), but there are key weaknesses (e.g., it describes incremental work), and it can significantly benefit from another round of revision. However, I won't object to accepting it if my co-reviewers champion it.

**Missing References:**

- BERT Models for Arabic Text Classification: A Systematic Review (Ali Saleh Alammary, 2022)
- Benchmarking Transformer-based Language Models for Arabic Sentiment and Sarcasm Detection (Ibrahim Abu Farha and Walid Magdy, 2021).

Many more references in literature for evaluating LLMs models for Arabic tasks and others for evaluating ChatGPT on non-English languages.

**Paper Topic And Main Contributions:**

This study aims to assess ChatGPT's capabilities in Arabic and its dialects. The research involves a comprehensive evaluation, including both automated and human assessments, across a broad spectrum of language understanding and generation tasks (44 tasks) on diverse datasets (60 datasets).

The focus is on comparing ChatGPT's performance in Arabic with smaller models that have been fine-tuned specifically for Arabic. The findings reveal that, despite its success in English, ChatGPT consistently falls short when compared to these monolingual fine-tuned models. A detailed comparison between ChatGPT and GPT-4 in Modern Standard Arabic and Dialectal Arabic further highlights limitations in handling Arabic dialects.

**Reasons To Accept:**

1- The paper addresses a significant gap in NLP research by investigating the performance of ChatGPT on Arabic and its dialects.

2- The paper goes beyond simple evaluation and includes a comparative analysis with smaller, fine-tuned Arabic models. This comparison helps highlight the specific strengths and weaknesses of ChatGPT in handling Arabic tasks.

3- The paper establishes a benchmark for evaluating ChatGPT's performance on Arabic tasks, which can serve as a reference for future studies.

**Reasons To Reject:**

1- The paper mentions that there is limited research on evaluating ChatGPT's performance on languages other than English, but it does not sufficiently integrate or discuss existing studies on NLP in non-English languages.

2- While the paper compares ChatGPT's performance with smaller fine-tuned models, it could benefit from more in-depth analyses of the differences in capabilities between the models. For example, specific tasks where ChatGPT falls short compared to dedicated models could be discussed in greater detail.

3- The paper states, "We randomly sample from each respective training dataset." Does this sampling process consider the class distribution for certain tasks? I believe further clarification is needed regarding the experimental setups. Additionally, the authors assert that MARBERT achieves the best results in the literature, but they have not provided sufficient references to support this claim.

4- The paper appears quite condensed. As a result, I would prefer a more comprehensive evaluation focused on a few tasks, delving deeply (evaluating ChatGPT and making comparative analysis with fine-tuned models for Arabic), rather than a broad approach across 44 tasks with concise explanations.

5- The paper lacks a discussion on the interpretability of ChatGPT's outputs. Providing insights into why the model may perform differently on Arabic and its dialects, for example by highlighting challenging Arabic features, could enhance the paper's depth.

**Reproducibility:**

3: Could reproduce the results with some difficulty. The settings of parameters are underspecified or subjectively determined; the training/evaluation data are not widely available.

**Reviewer Confidence:**

3: Pretty sure, but there's a chance I missed something. Although I have a good feel for this area in general, I did not carefully check the paper's details, e.g., the math, experimental design, or novelty.

---

> ### Author Rebuttal · Authors · 2023-08-29
>
> Dear Reviewer,
>
> Thank you for your insightful review. We are glad that you find our paper has addressed a significant research gap by investigating ChatGPT’s performance on Arabic NLP. Please find our response to your comments below.
>
> **W1. The paper mentions that there is limited research on evaluating ChatGPT's performance on languages other than English, but it does not sufficiently integrate or discuss existing studies on NLP in non-English languages.**
>
> Considering the number of languages ChatGPT has not been evaluated on at the time of submission, we believe that our claim that there is limited research on non-english remains true. The research gap is even more severe when it comes to languages with a wide range of dialects such as Arabic. In our paper, we cite a total of ten works related to evaluation of LLMs such as ChatGPT and BLOOMZ on non-English, as follows:
>
> First, we cover six notable works that evaluate ChatGPT on multilingual tasks (please see “Performance on Multilingual Tasks”, Section 2, lines 143-172). For example, we cite [1, 2] in which the authors evaluate ChatGPT/GPT-4 on medical licensing exams in Chinese and Japanese, respectively. Additionally, we include four more references particularly related to machine translation tasks involving non-English in Appendix A.1.
>
> That being said, since our work focuses on evaluating ChatGPT and other LLMs on Arabic, we also cite a number of previous work on Arabic NLP. This includes, for example, sources of our downstream datasets.
>
> Regardless, even though our paper has more than four pages of references, there is a chance that we may have missed some important relevant work and we will extend our literature review in the camera ready. For example, we will cover works that evaluate smaller models dedicated to Arabic.
>
>
> **W2. While the paper compares ChatGPT's performance with smaller fine-tuned models, it could benefit from more in-depth analyses of the differences in capabilities between the models. For example, specific tasks where ChatGPT falls short compared to dedicated models could be discussed in greater detail.**
>
> We believe this is a good suggestion and we will enhance our analysis about the differences in capabilities between the models. Meanwhile, in the analysis we provided we have offered several types of meaningful comparisons including on the differences between models. We summarize these here:
>
> We find that ChatGPT (and BLOOMZ) falls short compared to smaller finetuned models on the majority of the NLU and the NLG tasks (30 out of 35 tasks; Table 1 and Table 2), revealing the weakness of ChatGPT (and BLOOMZ) on Arabic (lines 106-109). We offer an analysis of this finding across the different tasks we consider. For example, we thoroughly discuss the limitation of ChatGPT on some sensitive tasks like toxicity detection (Appendix D), showing that ChatGPT exhibits higher false-toxicity rate. We further analyze tasks where ChatGPT indeed outperforms the dedicated models. For example, for WSD task, we suspect the reason is anisotropy (lines 352-355) and discuss this in detail in Appendix C. For code-switched translation tasks where ChatGPT also outperforms the smaller finetuned model, we point out that ChatGPT was likely exposed to other high resource languages like French during the pretraining and hence can perform well on code-switching.
>
> In addition, we provide a comparative analysis between the large language models and the smaller finetuned models across different Arabic varieties. For example, we show that limitations of the larger models such as ChatGPT and GPT-4 are more acute for Arabic dialects (rather than MSA) (lines 487-490). This can be directly correlated to availability of dialectal  pretraining data (lines 501-505) that were included in these LLMs. Since the smaller models are computationally less costly, we show that they can be easily finetuned on dialectal data to acquire high performance.
>
> Above all, we would also like to emphasize that it is difficult to provide a conclusive analysis of the outcomes of closed models such as ChatGPT due to lack of knowledge as to how these models are trained and what datasets have been used during their pretraining.
>
>
> **W3. The paper states, "We randomly sample from each respective training dataset." Does this sampling process consider the class distribution for certain tasks? I believe further clarification is needed regarding the experimental setups. Additionally, the authors assert that MARBERT achieves the best results in the literature, but they have not provided sufficient references to support this claim.**
>
> We thank the reviewer for this comment. We are providing the class distribution (in percentage) of the classification tasks (except for Dialect-Country where the number of classes is higher than the number of shots).
>
> | task (full dataset) | class_0 | class_1 | class_2 | class_3 | class_4 | class_5 | class_6 | class_7 |
> |--|--|--|--|--|--|--|--|--|
> | dialect-binary | 59.31 | 40.69 |  |  |  |  |  |  |
> | machine-gen | 49.87 | 50.13 |  |  |  |  |  |  |
> | dialect-region | 31.0 | 0.07 | 44.17 | 24.76 |  |  |  |  |
> | abusive | 8.02 | 29.19 | 62.79 |  |  |  |  |  |
> | hate speech | 5.12 | 94.88 |  |  |  |  |  |  |
> | offensive | 79.95 | 20.05 |  |  |  |  |  |  |
> | irony | 48.03 | 51.97 |  |  |  |  |  |  |
> | sarcasm | 15.91 | 84.09 |  |  |  |  |  |  |
> | dangerous | 71.16 | 28.84 |  |  |  |  |  |  |
> | adult | 11.86 | 88.14 |  |  |  |  |  |  |
> | gender | 53.31 | 46.69 |  |  |  |  |  |  |
> | age | 33.98 | 30.26 | 35.76 |  |  |  |  |  |
> | claim | 67.5 | 32.5 |  |  |  |  |  |  |
> | emotion | 29.62 | 1.8 | 13.75 | 15.13 | 13.3 | 2.43 | 8.44 | 15.53 |
> | sentiment | 55.04 | 31.9 | 13.06 |  |  |  |  |  |
> | mq2q | 0.01 | 44.76 | 55.23 |  |  |  |  |  |
> | stance | 34.05 | 63.57 | 2.38 |  |  |  |  |  |
> | xnli | 33.05 | 33.69 | 33.26 |  |  |  |  |  |
> | wsd | 49.99 | 50.01 |  |  |  |  |  |  |
>
> We are further providing the class distribution of the 10-shot training samples that we randomly selected.
>
> | task (10-shot) | class_0 | class_1 | class_2 | class_3 | class_4 | class_5 | class_6 | class_7 |
> |--|--|--|--|--|--|--|--|--|
> | dialect-binary | 60.0 | 40.0 |  |  |  |  |  |  |
> | machine-gen | 50.0 | 50.0 |  |  |  |  |  |  |
> | dialect-region | 40.0 | 0.0 | 40.0 | 20.0 |  |  |  |  |
> | abusive | 10.0 | 20.0 | 70.0 |  |  |  |  |  |
> | hate speech | 10.0 | 90.0 |  |  |  |  |  |  |
> | offensive | 80.0 | 20.0 |  |  |  |  |  |  |
> | irony | 40.0 | 60.0 |  |  |  |  |  |  |
> | sarcasm | 20.0 | 80.0 |  |  |  |  |  |  |
> | dangerous | 70.0 | 30.0 |  |  |  |  |  |  |
> | adult | 10.0 | 90.0 |  |  |  |  |  |  |
> | gender | 60.0 | 40.0 |  |  |  |  |  |  |
> | age | 40.0 | 30.0 | 40.0 |  |  |  |  |  |
> | claim | 80.0 | 20.0 |  |  |  |  |  |  |
> | emotion | 50.0 | 0.0 | 10.0 | 10.0 | 20.0 | 0.0 | 10.0 | 0.0 |
> | sentiment | 60.0 | 20.0 | 10.0 |  |  |  |  |  |
> | mq2q | 0.0 | 30.0 | 70.0 |  |  |  |  |  |
> | stance | 40.0 | 50.0 | 10.0 |  |  |  |  |  |
> | xnli | 40.0 | 30.0 | 30.0 |  |  |  |  |  |
> | wsd | 50.0 | 50.0 |  |  |  |  |  |  |
>
>
> As evident from the tables, the class distribution of the few-shot samples is reasonably aligned with the corresponding tasks. Therefore, the sampling process indeed considers the class distribution of the respective tasks. We will add this in the appendix of the paper.
>
> Regarding the performance of MARBERT, we referred to the original paper (line 248), where the authors claim the superiority of the model. We will put the same reference again along with line 251.
>
>
> **W4. The paper appears quite condensed. As a result, I would prefer a more comprehensive evaluation focused on a few tasks, delving deeply (evaluating ChatGPT and making comparative analysis with fine-tuned models for Arabic), rather than a broad approach across 44 tasks with concise explanations.**
>
> While we agree to the benefit of focusing on fewer tasks, we also see the privilege of providing a comprehensive evaluation across a large number of tasks that represent Arabic at scale.
>
> That being said, we also methodically group tasks into either natural language understanding (NLU) or natural language generation (NLG). This breakdown immediately demonstrates capabilities of the models across these two broad categories. We also provide analysis on some selected tasks like toxicity detection (false toxicity in Appendix D), WSD (anisotropy in Appendix C). One challenge we have is page limits, and we wanted to strike a balance between what we put in the main body of the paper and a reasonable length for appendixes. However, with an extra page for the camera-ready, we will provide a more in-depth analysis of ChatGPT on selected tasks following the suggestion. For example, we have further probed ChatGPT’s code-switching ability using diagnostic test cases. Specifically, we manually evaluated ChatGPT’s capability of English mixed MSA, Egyptian, and Moroccan code-switched translation generation from plain English text. We ask two annotators who are fluent on the respective varieties (as well as English), to evaluate a the diagnostic dataset based on fluency (defined as how fluent the translated text is), faithfulness (defined as how semantically close the translated text is to the source text), and code-switching ability (defined as how accurately the translated text includes the code-switch scripts). We provide clear instructions to annotators about the task. We present our result (in percentage) in the table below, and will provide the details of the full analysis in the camera-ready.
>
>
> | Annotator | CST | fluency (A/B/C/D) | faithfulness (A/B/C/D) | code-switching (A/B/C/D) |
> |--|--|--|--|--|
> | Annotator 1 | En->MSA_En | 80/10/10/0 | 80/20/0/0 | 0/40/0/60 |
> |  | En->Egy_En | 70/10/20/0 | 100/0/0/0 | 20/30/0/50 |
> |  | En->Mor_En | 10/40/50/0 | 90/0/10/0 | 20/30/0/50 |
> |  | ***Avg*** | 53.3/20/26.7/0 | 90/6.7/3.3/0 | 13.3/33.3/0/53.4 |
> | Annotator 2 | En->MSA_En | 30/70/0/0 | 90/0/10/0 | 0/30/0/70 |
> |  | En->Egy_En | 50/40/10/0 | 90/10/0/0 | 0/40/0/60 |
> |  | En->Mor_En | 20/40/40/0 | 100/0/0/0 | 20/30/0/50 |
> |  | ***Avg*** | 33.3/50/16.7 | 93.4/3.3/3.3/0 | 6.7/33.3/0/60 |
> | | ***Average*** | 43.3/35/21.7/0 | 91.7/5/3.3/0 | 10/33.3/0/56.7 |
>
>
>
> We find that ChatGPT produces fluent translations that are also semantically close (faithful) to the source text. However, ChatGPT struggles to produce code-switched text and generates mostly in Arabic script. Interestingly, this issue is more prevalent for MSA than Egyptian and Moroccan. We hypothesize that the tokens in the source English text are very common in MSA. As a result, the model does not seem ‘willing’ to code-switch the words into English.
>
> **W5. The paper lacks a discussion on the interpretability of ChatGPT's outputs. Providing insights into why the model may perform differently on Arabic and its dialects, for example by highlighting challenging Arabic features, could enhance the paper's depth.**
>
>
> We have carried out an analysis of ChatGPT’s ability to translate from different varieties of Arabic into English. In this analysis, we observe a number of challenges that we believe are unique to Arabic. This involves a discrepancy between the model’s ability to translate from MSA and their ability to translate from Arabic dialects such as Egyptian or Moroccan. This discrepancy cuts across several linguistic categories and involves a discussion of topics such as translation of endearment expressions; multi-word expressions, idioms, and proverbs; negation; sub-token-level code-switching; and dialectal variants of MSA lexica. Across all of these areas, we observe challenges faced by ChatGPT and we will include this analysis in our camera-ready version of the paper. We believe this analysis will offer further insights and additional depth as to the interpretability of ChatGPT's outputs vis-a-vis Arabic and its dialects.
>
> In addition to this, we would like to emphasize that our work provides an extensive systematic evaluation (line 67, line 106) that shows how ChatGPT fares on Arabic. Our work demonstrates that ChatGPT performs better on MSA than DA data (Section 8), because the model has likely seen more MSA data (lines 487-489). We further probe the performance variation across the different dialects and conclude that the potential reason is due to the unavailability of dialectal pretraining data (lines 501-505).
>
> Moreover, we conduct human evaluation for those tasks where automatic metrics fail to capture overall quality of the output (Section 9). Our evaluation framework to an extent can be used to interpret output of ChatGPT and other LLMs. For example, we state that rating-A is when the output is acceptable without any issue. While rating-B and rating-C are when the outputs have minor to major hallucination, toxicity, grammatical issues. Rating-D is when the output is not a response to the prompt. Again, we will augment these already offered insights with our new analysis we describe above.
>
>
> **Missing References**
>
> *M1. BERT Models for Arabic Text Classification: A Systematic Review (Ali Saleh Alammary, 2022)*
>
> *M2. Benchmarking Transformer-based Language Models for Arabic Sentiment and Sarcasm Detection (Ibrahim Abu Farha and Walid Magdy, 2021).*
>
> *M3. Many more references in literature for evaluating LLMs models for Arabic tasks and others for evaluating ChatGPT on non-English languages.*
>
> We thank the reviewer for providing these references. We will add the mentioned references and other relevant literature to our paper.
>
>
> ***References***
>
> [1] Wu, Jiageng, et al. "Qualifying Chinese Medical Licensing Examination with Knowledge Enhanced Generative Pre-training Model." arXiv preprint arXiv:2305.10163 (2023).
>
> [2]. Kasai, Jungo, et al. "Evaluating gpt-4 and chatgpt on japanese medical licensing examinations." arXiv preprint arXiv:2303.18027 (2023).

---

### Meta-Review · Area_Chair_3GXJ · 2023-09-18

**Recommendation:** 5

**Metareview:**

This paper presents a large scale evaluation of the performance of ChatGPT on Modern standard Arabic and other Arabic dialects. The evaluation cover 44 distint taks and 60 datasets.

The reviewers found this to be a valuable contribution in terms of benchmarking ChatGPT performance on Arabic tasks. There are a few concerns about the paper is too condensed giving the number of evaluation tasks and datasets. It would be great if the authors add more insights on the evaluation in the paper or appendix.

---

### Decision · Program_Chairs · 2023-10-07

**Decision:**

Accept-Main

**Comment:**

This paper presents a large scale evaluation of the performance of ChatGPT on Modern standard Arabic and other Arabic dialects. The evaluation cover 44 distint taks and 60 datasets.

The reviewers found this to be a valuable contribution in terms of benchmarking ChatGPT performance on Arabic tasks. There are a few concerns about the paper is too condensed giving the number of evaluation tasks and datasets. It would be great if the authors add more insights on the evaluation in the paper or appendix.